# GraphP-FL: Personalized Federated Graph Learning via Dynamic Structure Awareness and Fisher Information Elastic Alignment

Haoyu Chen [1 2]   Zening Zhao [1 2]   Jinsong Wang [1 2 3]   Kai Shi [1 2]   Zongpu Wei [1 2]   Jianhao Li [1 2]

## Abstract

Federated Graph Learning (FGL) enables distributed clients to collaboratively train graph neural networks while strictly preserving data privacy. However, existing FGL methods implicitly assume the reliability of local graph structures and lack elastic awareness of parameter importance during model aggregation, leading to representation degradation under topological noise and catastrophic forgetting caused by model drift. To address these challenges, we propose GraphP-FL, a general personalized FGL framework. (1) Specifically, we design a self-supervised dynamic topology reconstruction mechanism on the client side. This mechanism mines implicit dependencies to adaptively rectify noisy topologies, effectively suppressing topological noise propagation and capturing precise structural relationships for high-quality representations. (2) Additionally, we introduce a Fisher-based Elastic Parameter Alignment (FRPA) algorithm. FRPA imposes anisotropic regularization constraints in the parameter space to precisely quantify parameter importance, enabling the model to strictly preserve critical local knowledge while flexibly aligning with the global model, thus effectively overcoming catastrophic forgetting. Extensive experiments on seven benchmarks (including biochemical molecules, social networks, and large-scale encrypted traffic) demonstrate that GraphP-FL significantly outperforms state-of-the-art methods, improving accuracy by up to 8.6% while exhibiting superior generalization and robustness.

[1]Tianjin University of Technology [2]Tianjin Key Laboratory of Intelligence Computing and Novel Software Technology [3]Tianjin University of Technology and Education. Correspondence to: Zening Zhao <znzhao@email.tjut.edu.cn>, Jinsong Wang <jswang@tjut.edu.cn>.

*Proceedings of the 43rd International Conference on Machine Learning*, Seoul, South Korea. PMLR 306, 2026. Copyright 2026 by the author(s).

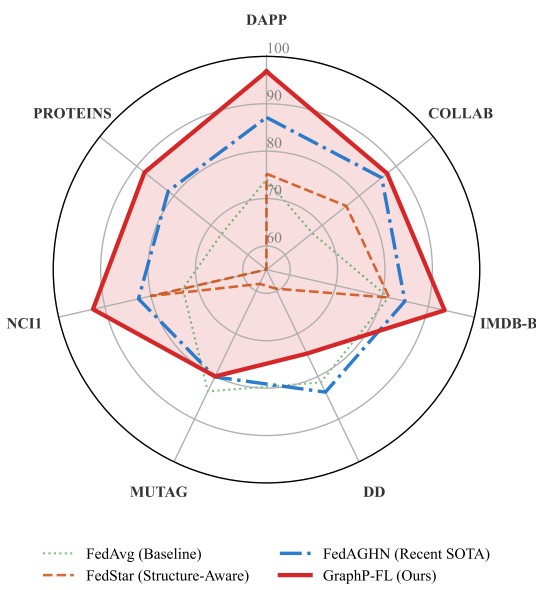

*Figure 1.* Comparison of test accuracy on seven heterogeneous graph datasets. The red area illustrates the performance envelope of **GraphP-FL**, which significantly outperforms existing SOTA methods.

## 1. Introduction

As the volume of graph data generated by distributed devices grows exponentially, Federated Graph Learning (FGL) has emerged as the standard paradigm (Li et al., 2024b) for breaking down "data silos" while complying with privacy regulations such as GDPR (Yang et al., 2019). By collaboratively training Graph Neural Networks (GNNs)—such as Graph Transformers (Kong et al., 2024), Graph Mambas (Wang et al., 2024), and Diffusion-based GNNs (Zhou et al., 2024a)—without sharing raw graph topologies, FGL has demonstrated immense potential in domains including biochemical molecule discovery, financial risk control, and recommender systems (Li et al., 2025d). However, most existing studies operate under idealized experimental settings, assuming that clients possess high-quality graph structures and relatively uniform data distributions (Huang et al., 2025; Xu et al., 2025). In contrast, real-world scenarios are of-

ten plagued by a *dual heterogeneity* challenge (Fu et al., 2025a), characterized by the coupling of latent structural noise and statistical distribution drift. This coupling directly precipitates collaboration failure, potentially rendering the performance of the federated model inferior even to that of local models (Li et al., 2024b; Yang et al., 2024b).

First, existing methods generally harbor unrealistic assumptions regarding the completeness of local graph structures. In real-world scenarios, due to limitations in data collection or the injection of privacy-preserving noise, client-held graph topologies are often noisy, sparse, or even task-irrelevant (Wang et al., 2026). Traditional FGL frameworks typically perform message passing directly on these low-quality graphs. This approach overlooks the fact that high-quality adjacency is a prerequisite for effective GNN learning; erroneous edges propagate misleading features, leading to local representation degeneration. Exacerbating this issue is the severe label scarcity dilemma faced by clients (Chen et al., 2024). Relying solely on sparse supervision signals makes it difficult to effectively guide complex graph structure reconstruction. Although centralized approaches like GraphCroc (Duan et al., 2024) and LGD (Zhou et al., 2024b) attempt to optimize topology, they are prone to overfitting noise in federated settings (Fatemi et al., 2021). Therefore, mining intrinsic data signals to adaptively reconstruct discriminative latent structures under supervision-starved local environments remains an urgent challenge.

Second, catastrophic forgetting induced by Non-IID data distributions remains an unresolved hurdle in collaborative optimization. Due to user preferences or geographic discrepancies, local optima often deviate significantly from the global optimum, a phenomenon known as model drift (Fanì et al., 2024). While pioneering works like FedProx (Li et al., 2020) introduce proximal terms to constrain local updates, their isotropic constraint strategies lack fine-grained awareness of parameter functional importance. By imposing indiscriminate constraints on all parameters, they force the model to forget distinct local knowledge during alignment with the global model (Setayesh et al., 2022). Furthermore, existing anti-forgetting mechanisms often ignore the strong coupling between parameter importance and graph topology (Tan et al., 2023). In scenarios with noisy structures and heterogeneous distributions, how to elastically protect local personalized knowledge based on structural sensitivity while simultaneously correcting the topology remains an open problem.

To address these challenges, we propose **GraphP-FL**, a general personalized federated graph learning framework. Specifically, we first introduce a self-supervised enhanced dynamic structure learning mechanism. Utilizing Graph Contrastive Learning (GCL) (You et al., 2020; Tan et al., 2024; He et al., 2024) as a structural regularizer, this mech-

anism mines multi-scale dependencies intrinsic to the graph data under the constraint of scarce supervision. This allows for the adaptive reconstruction of local graph topologies, effectively suppressing the propagation of topological noise and significantly enhancing the discriminability and robustness of graph representations. Inspired by continual learning (Kirkpatrick et al., 2017; Li et al., 2025b), we introduce a Fisher-based Elastic Parameter Alignment (FRPA) mechanism to mitigate catastrophic forgetting. FRPA utilizes the Fisher Information Matrix to precisely quantify parameter functional importance (i.e., loss curvature) and imposes anisotropic constraints. This drives non-critical parameters to flexibly align with the global model while rigorously protecting core parameters associated with local knowledge.As illustrated in Figure 1, these mechanisms enable GraphP-FL to establish a superior performance envelope across diverse heterogeneous scenarios, significantly outperforming state-of-the-art baselines.

Our main contributions are summarized as follows:

**(1) Personalized FGL Framework:** We propose GraphP-FL, which synergizes structure awareness in the data space with elastic alignment in the parameter space to effectively mitigate representation degeneration and catastrophic forgetting under dual heterogeneity.

**(2) Dynamic Structure Learning Mechanism:** We leverage implicit signals from graph contrastive learning to guide dynamic topology reconstruction under label scarcity. This mechanism adaptively filters redundancy edges, fundamentally enhancing data-level noise resilience and discriminability.

**(3) Fisher-based Elastic Parameter Alignment (FRPA):** To overcome catastrophic forgetting induced by model drift, we innovatively utilize Fisher information to construct anisotropic parameter constraints. This allows the local model to align with the global model while strictly preserving core parameters, effectively overcoming forgetting.

**(4) Extensive Experiments:** Comprehensive evaluation on seven benchmarks across biochemical, social, and encrypted traffic domains demonstrates that GraphP-FL outperforms SOTA methods by up to 8.6%, exhibiting superior generalization and robustness. Ablation studies further validate the necessity of each module.

## 2. Related Work

### 2.1. Graph Structure Learning and Self-Supervised Graph Representation

Existing research leveraging Graph Structure Learning (GSL) and Self-Supervised Learning (SSL) primarily follows two pathways: explicit structure reconstruction and

implicit representation enhancement. Structure reconstruction focuses on correcting topological biases using attribute features or multi-expert mechanisms. For instance, EF-GAE (Li et al., 2024a) fuses multi-modal features via autoencoders to eliminate redundant connections, while DADC (Li et al., 2025a) and MEGC (Fu et al., 2025b) design domain-specific denoising and pruning mechanisms. Representation enhancement methods mine implicit dependencies intrinsic to data by maximizing multi-view mutual information. GRACE (Tan et al., 2024) and its successors (Zhu et al., 2021; Thakoor et al., 2021) established the contrastive paradigm based on data augmentation. Subsequent works further introduced cross-view semantic alignment (Zhang & Bao, 2025; Wang et al., 2025a; Liang et al., 2025), long-range dependency capture (Li et al., 2025c), and asymmetric spectral augmentation (Liu et al., 2025) to overcome label scarcity.

However, these centralized methods rely heavily on global topological views and abundant supervision unavailable under data isolation, and are confined to feature-level enhancements that fail to explicitly reconstruct noisy structures.

### 2.2. Personalized Federated Learning and Anti-Forgetting Mechanisms

Personalized Federated Learning (PFL) offers parameter-level solutions to address statistical heterogeneity. Existing research primarily mitigates catastrophic forgetting through two pathways: parameter decoupling and collaborative optimization. Parameter decoupling strategies focus on decomposing the model into a shared backbone and a personalized head, mitigating knowledge interference via physical isolation(Liang et al., 2020; Collins et al., 2021).FedECP (Fu et al., 2025c) and FedMCA (Zheng et al., 2025) employed split learning strategies to further isolate shared and private parameters. In parallel, Collaborative Optimization balances global collaboration with local adaptation. Ditto (Li et al., 2021b) and MOON (Li et al., 2021a) constrain local update directions via regularization and contrastive loss, respectively, while pFedMLKD (Xia et al., 2025) and FedBM (Ping et al., 2025) transfer global paradigms via knowledge distillation.

Despite these advancements, general PFL methods overlook the strong coupling between parameters and topology in GNNs, while coarse-grained collaborative optimization fails to perceive local parameter curvature required for dual heterogeneity, resulting in suboptimal updates that cannot prevent catastrophic forgetting.

### 2.3. Federated Graph Learning

In the field of Federated Graph Learning (FGL), existing works primarily address dual heterogeneity through distinct mechanisms. **Regarding Statistical Heterogeneity,** GCFL+ (Xie et al., 2021) and PFL-Graph (Chen et al., 2022) attempt to mitigate distribution conflicts across clients via gradient-based dynamic clustering and functional mapping mechanisms, respectively. Furthermore, HarmoFGL (Yan et al., 2026) specifically tackles feature deviations by utilizing implicit feature crossing to disentangle client-universal and client-specific interactions. **Regarding Structural Heterogeneity and Completion,** FedSage+ (Zhang et al., 2021) and FedNGKD (Wang et al., 2025b) utilize generative models to fill missing edges across clients, while FedStar (Tan et al., 2023) and Position-Aware FGL (Dai et al., 2025) focus on aligning cross-domain topologies through shared structural embeddings or positional encodings. FedProto (Tan et al., 2022) proposes a prototype-based cross-client knowledge alignment mechanism. Additionally, PerFedGT (Jia et al., 2025) and FedHGCL (Wu et al., 2025) further introduce Transformer architectures and contrastive learning paradigms to enhance representation capabilities in complex scenarios.

Existing methods largely neglect explicit structure denoising and lack elastic aggregation strategies to balance knowledge retention under heterogeneity. Although Yang et al. (2024a) utilize the Fisher Information Matrix (FIM) trace for weighting, their model-level approach remains too coarse-grained, failing to capture both the **irregular topological dependencies** and the fine-grained parameter importance essential for robust collaboration (Li et al., 2025b).

## 3. Preliminaries and Problem Formulation

Following the notation definitions, we formulate the personalized FGL problem from a probabilistic perspective. Specifically, we model structural noise and heterogeneity as latent variable inference and Riemannian prior constraints, respectively, laying the theoretical foundation for GraphP-FL.

### 3.1. Notation

Consider a federated system consisting of $N$ clients $\mathcal{C} = \{C_1, C_2, \ldots, C_N\}$. Each client $C_i$ holds a private graph dataset $\mathcal{D}_i = \{(G_j, y_j)\}_{j=1}^{|\mathcal{D}_i|}$, where $|\mathcal{D}_i|$ denotes the number of samples. Each graph instance is represented as $G = (\mathcal{V}, \mathcal{E}, X, A)$, where $\mathcal{V}$ is the node set, $\mathcal{E}$ is the edge set, $X \in \mathbb{R}^{M \times d}$ is the node feature matrix, and $A \in \{0,1\}^{M \times M}$ is the adjacency matrix. We define the graph classification task as learning a mapping function $f_\theta : \mathcal{G} \to \mathcal{Y}$, where $\theta$ represents the model parameters and $\mathcal{Y} = \{1, \ldots, K\}$ is the label set. To quantify the importance of the parameter space, we introduce the Fisher Information Matrix (FIM), denoted as $F$. For a parameter vector $\theta$, its diagonal approximation is defined as $F = \text{diag}(F_1, \ldots, F_{|\theta|})$, which defines a Riemannian metric on the parameter space.

## 3.2. Problem Definition

Based on the notation defined above, in a standard federated learning setting, each client $C_i$ aims to minimize the empirical risk on its local dataset $\mathcal{D}_i$. Traditional optimization objectives typically assume that the graph structure is complete and the parameter space is a flat Euclidean space, *i.e.*, $\min_{\theta_i} \mathbb{E}_{(G,y)\sim\mathcal{D}_i}[\mathcal{L}(\theta_i; G, y)]$, where $\mathcal{L}$ is the standard supervised classification loss. However, in personalized federated graph learning, training difficulties and performance degradation often arise from the coupling of **dual heterogeneity**:

- **Statistical Heterogeneity:** We assume significant differences in the marginal label distributions across clients, *i.e.*, $P_i(Y) \neq P_j(Y)$. This distribution shift causes the optimal solutions of local likelihood functions to diverge on the parameter manifold, implying that simple parameter averaging ignores the geometric properties of different client parameter distributions.

- **Structural Heterogeneity:** We posit that the observed adjacency matrix $A_{obs}$ is not the absolute ground truth but rather a result sampled from the latent ground-truth structure $A_{gt}$ through a noisy channel, following the conditional probability $A_{obs} \sim P(A|A_{gt})$. Direct training on noisy observations leads to overfitting task-irrelevant topological patterns.

To achieve robust personalized FGL under these challenges, we formulate the optimization objective as a Maximum A Posteriori (MAP) estimation problem. Specifically, we introduce an Information Bottleneck-based structural prior to handle noise and a Fisher Information-based parameter prior to handle drift. Consequently, the final personalized loss function of GraphP-FL is defined as:

$$\min_{\{\theta_i\},\tilde{A}} \sum_{i=1}^{N} \left[ \underbrace{\mathcal{L}_{\text{sup}}(\theta_i; \tilde{A}, \mathcal{D}_i)}_{\text{Task Risk}} + \lambda_1 \underbrace{\mathcal{L}_{\text{IB}}(\tilde{A}; A_{\text{obs}})}_{\text{Struct. IB}} \right.$$
$$\left. + \lambda_2 \underbrace{\mathcal{R}_{\text{Riemann}}(\theta_i, \theta_g)}_{\text{Elastic Align.}} \right] \tag{1}$$

where:

- $\mathcal{L}_{\text{sup}}(\theta_i; \tilde{A}, \mathcal{D}_i)$ represents the supervised classification loss on the inferred structure $\tilde{A}$. From a probabilistic view, this term maximizes the likelihood $P(Y|X, \tilde{A}; \theta_i)$, ensuring the model fits the local-specific data distribution.

- $\mathcal{L}_{\text{IB}}(\tilde{A}; A_{\text{obs}})$ denotes the Information Bottleneck-based structural regularization. We infer the latent $A_{gt}$ by finding a minimal sufficient statistic $\tilde{A}$. Theoretically, $\mathcal{L}_{\text{sup}}$ encourages sufficiency (maximizing

$I(\tilde{A}; Y)$), while this term enforces compression (minimizing $I(\tilde{A}; A_{obs})$) to filter out noise.

- $\mathcal{R}_{\text{Riemann}}(\theta_i, \theta_g)$ is the Riemannian manifold elastic regularization. Given the non-Euclidean parameter space caused by statistical heterogeneity, we use the global Fisher Information Matrix $F_g$ to define the Mahalanobis distance: $\mathcal{R}_{\text{Riemann}}(\theta_i, \theta_g) := \frac{1}{2}(\theta_i - \theta_g)^\top F_g(\theta_i - \theta_g)$. Ideally, it forces strict alignment with the global consensus in high-curvature directions while allowing personalized adaptation in flat directions. $\lambda_1, \lambda_2$ are balancing hyperparameters.

# 4. Method

## 4.1. Overall Framework

The core of GraphP-FL is to establish an elastic collaboration mechanism (Figure 2). On the client side, we deploy a self-supervised dynamic graph encoder that leverages the Information Bottleneck principle to actively rectify noisy structures, extracting robust minimal sufficient statistics. On the server side, a Fisher Information-based knowledge base guides curvature-aware weighted aggregation. This mechanism replaces naive averaging with anisotropic personalized alignment, achieving an optimal balance between anti-forgetting and personalization.

## 4.2. Local Dynamic Topology Reconstruction via Self-Supervision

Before executing structure learning, clients first instantiate heterogeneous local data into a unified graph structure format. For natural graph data (e.g., biochemical molecules), we retain their original topology. For non-graph stream data (e.g., encrypted traffic), we employ a specialized graph construction method that maps continuous packet sequences into traffic interaction graphs containing temporal dependencies, thereby transforming unstructured temporal features into spatial structures $G_{obs}$ processable by GNNs (see Appendix A for details).

**(a) Adaptive Feature Denoising via Variational Information Bottleneck.** To materialize the structural prior proposed in Section 3, we design a dynamic denoising module based on the Variational Information Bottleneck (VIB). Our goal is to learn a latent denoised representation $Z$ that maximizes the mutual information with the prediction target $I(Z; Y)$ while minimizing the mutual information with the noisy input $I(Z; G_{obs})$. Since mutual information for high-dimensional graph data is intractable, we employ variational inference by introducing a parameterized variational distribution $p_\Phi(Z|G_{obs})$ to approximate the true posterior (see Appendix B for proofs). We instantiate this variational distribution as a deterministic soft masking mechanism. Specifically, we utilize a lightweight structure learner $f_{mask}(\cdot)$ to

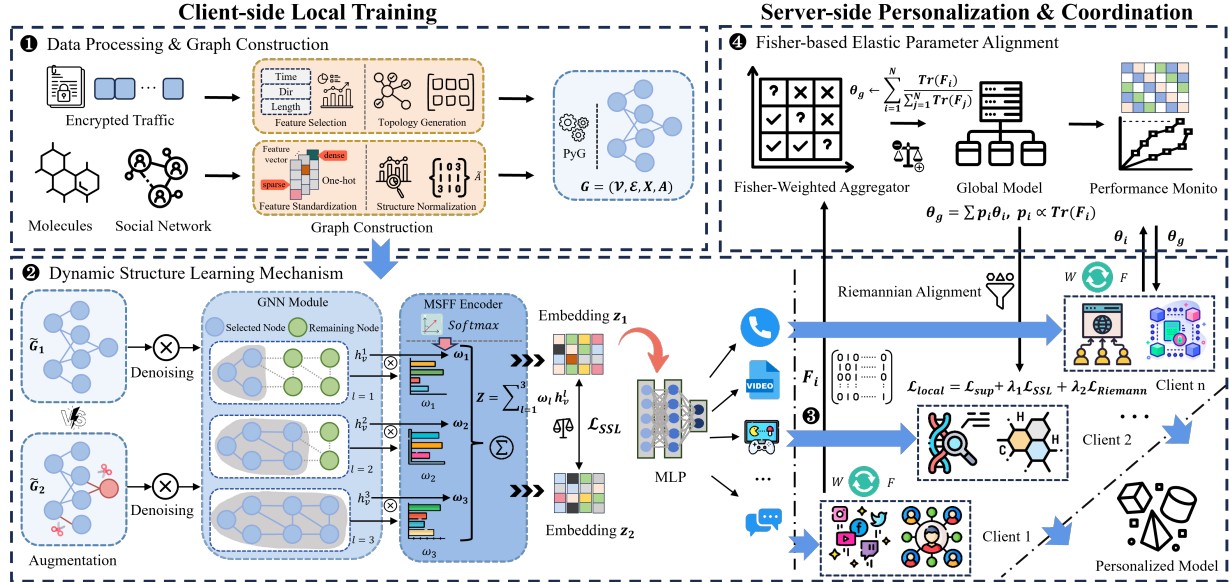

*Figure 2.* **The overall framework of GraphP-FL.** The training process consists of four key stages: **(1) Graph Construction:** Clients construct graph structures based on raw input data. **(2) Dynamic Structure Learning Mechanism (Sec. 4.2):** Adopts an adaptive denoising module and MSFF encoder to extract robust node representations under a dual-stream supervision mechanism. **(3) Fisher Estimation:** Calculates the Fisher Trace using gradient information to quantify model reliability. **(4) Fisher-based Elastic Parameter Alignment (Sec. 4.3):** The server executes curvature-based weighted aggregation and guides clients in Riemannian manifold-based elastic alignment to effectively mitigate catastrophic forgetting.

evaluate feature importance dimension-wise:

$$M = \sigma(W_2 \cdot \text{ReLU}(W_1 X + b_1) + b_2) \quad (2)$$

where $\sigma(\cdot)$ is the Sigmoid function mapping outputs to the $(0, 1)$ interval, representing the retention probability of feature dimensions. $W, b$ are learnable variational parameters. Based on the generated mask $M$, we perform soft pruning on the original features to obtain the latent representation $Z$:

$$Z = X \odot M \quad (3)$$

This mechanism constructs a differentiable information gate. By minimizing the Information Bottleneck objective, the model adaptively identifies and blocks task-irrelevant noise features. Since message passing in GNNs relies on feature interactions, feature denoising implicitly rectifies erroneous connection weights in the adjacency matrix, achieving robust reconstruction of $A_{obs}$ from the data foundation.

**(b) Multi-scale Representation Fusion and Self-supervised Collaborative Optimization.** To extract robust semantic representations from the latent denoised structure $\tilde{G}$ obtained in Section 4.2 (a) and address the scarcity of supervision signals in federated scenarios, we propose a collaborative optimization strategy fusing hierarchical encoding and self-supervised auxiliary tasks. Considering the structural heterogeneity of graph data, a single-scale view struggles to capture complete topological semantics. We adopt a multi-layer Graph Isomorphism Network (GIN) as

the backbone encoder and design a multi-scale context aggregation mechanism. Unlike using only the output of the last layer, we treat the representation $h_{\tilde{G}}^{(l)}$ of each layer as a structural view under different receptive fields and fuse them via an adaptive attention mechanism:

$$z = \sum_{l=1}^{L} \alpha_l \cdot \text{READOUT}(h_{\tilde{G}}^{(l)}),$$
$$\alpha_l = \frac{\exp(\text{Attn}(h_{\tilde{G}}^{(l)}))}{\sum_{k=1}^{L} \exp(\text{Attn}(h_{\tilde{G}}^{(k)}))} \quad (4)$$

This mechanism enables the model to adaptively assign higher weights to high-level features in noise-dense local regions, utilizing global context to smooth out residual local topological perturbations, further enhancing robustness against structural uncertainty. However, relying solely on sparse supervision signals is insufficient to guide high-quality latent structure inference. Thus, we introduce a self-supervised contrastive learning stream alongside the supervised stream. We construct two augmented views $\tilde{G}_1, \tilde{G}_2$, aiming to maximize the mutual information $I(z_1; z_2)$ of the same graph instance under different views. We adopt the NT-Xent loss as a lower bound estimate of mutual information:

$$\mathcal{L}_{SSL} = -\log \frac{\exp(\text{sim}(z_1, z_2)/\tau)}{\sum_{z' \in \{z_2\} \cup \mathcal{Z}_{neg}} \exp(\text{sim}(z_1, z')/\tau)} \quad (5)$$

From a probabilistic perspective, minimizing $\mathcal{L}_{SSL}$ maximizes the mutual information between structurally aug-

mented views, which effectively instantiates the information bottleneck constraint $\mathcal{L}_{IB}$ in Eq. (1). By forcing the encoder to ignore subtle topological changes remaining after denoising, it captures intrinsic, structurally invariant semantic information. This acts as a consistency prior that complements the masking mechanism in Section 4.2 (a). Therefore, combining variational denoising, self-supervised augmentation, and Fisher parameter alignment, the final local optimization objective for the client is formalized as:

$$
\mathcal{L}_{local} = \underbrace{\mathcal{L}_{sup}(\hat{y}, y)}_{\text{Task Risk}} + \lambda_1 \underbrace{\mathcal{L}_{SSL}}_{\text{Invariant Prior}}
$$
$$
+ \frac{\lambda_2}{2} \underbrace{(\theta_i - \theta_g)^\top F_g (\theta_i - \theta_g)}_{\text{Riemannian Constraint}} \tag{6}
$$

where $\mathcal{L}_{sup}$ is the task-related supervised loss. $\mathcal{L}_{SSL}$ instantiates the structural prior, ensuring representation invariance. The third term is the Riemannian regularization term based on Fisher Information, which utilizes the global Fisher matrix $F_g$ distributed by the server to constrain the local update direction, forcing the local model to align with the global model while protecting key knowledge(see derivation in Appendix C).

### 4.3. Fisher-based Elastic Collaboration Mechanism

Statistical heterogeneity causes the optimization trajectories of different clients to diverge, leading to model drift. To address catastrophic forgetting induced by drift, we establish an elastic collaboration mechanism based on Riemannian geometry on the server. The overall collaborative training process is summarized in Algorithm 1.

**(a) Elastic Parameter Alignment on Riemannian Manifold.** Traditional collaborative optimization methods like FedProx employ Euclidean distance $\|\theta - \theta_g\|^2$ for constraints, ignoring the sensitivity differences of different parameters to topological perturbations in GNNs. During the local training phase (Algorithm 1, Lines 6-12), to instantiate the Riemannian prior proposed in Section 3, clients utilize the global Fisher Information Matrix $F_g$ to define the geometric metric of the parameter space and impose Fisher Regularized Parameter Alignment (FRPA):

$$
\mathcal{R}_{\text{Riemann}}(\theta_i, \theta_g) := \frac{1}{2}(\theta_i - \theta_g)^\top F_g (\theta_i - \theta_g)
$$
$$
\approx \frac{1}{2} \sum_k F_{g,k} (\theta_{i,k} - \theta_{g,k})^2 \tag{7}
$$

where $F_g$ adopts a diagonal approximation to reduce computational complexity from $O(d^2)$ to $O(d)$. This regularization term essentially imposes anisotropic elastic constraints on the parameter manifold: $F_{g,k}$ acts as an importance coefficient for each parameter dimension. For high-curvature

---

**Algorithm 1** Fisher-based Elastic Collaboration

**Require:** Global rounds $T$, local epochs $E$, learning rate $\eta$;
        Local graph datasets $\{\mathcal{D}_i\}$ held by clients.
**Ensure:** Final global model $\theta_g$.
1: **Server Init:** $\theta_g^0$ random, Fisher Matrix $F_g^0 \leftarrow \mathbf{0}$.
2: **for** round $t = 0, \dots, T-1$ **do**
3:     Server broadcasts $\theta_g^t, F_g^t$ to active clients $\mathcal{S}_t$.
4:     **for** client $i \in \mathcal{S}_t$ in parallel **do**
5:        Init local model $\theta_i \leftarrow \theta_g^t$.
6:        *// Step 1: Structure-Aware Optimization*
7:        **for** epoch $e = 1, \dots, E$ **do**
8:           Sample batch $\mathcal{B} \sim \mathcal{D}_i$, get views $\tilde{G}_1, \tilde{G}_2$.
9:           Denoise (Eq. 2) and fuse features (Eq. 4).
10:          $\mathcal{L}_{batch} = \mathcal{L}_{sup} + \lambda_1 \mathcal{L}_{SSL} + \frac{\lambda_2}{2}(\theta_i - \theta_g^t)^\top F_g^t (\theta_i - \theta_g^t)$
11:          Update $\theta_i \leftarrow \theta_i - \eta \nabla \mathcal{L}_{batch}$.
12:        **end for**
13:        *// Step 2: Fisher Information Estimation*
14:        Compute Fisher diag $F_i$ and trace $I_i = \text{Tr}(F_i)$.
15:        Upload $\theta_i$, $F_i$, and $I_i$ to Server.
16:     **end for**
17:     *// Server-side: Weighted Aggregation*
18:     Receive updates; Calc weights $w_i = I_i / \sum_{j \in \mathcal{S}_t} I_j$.
19:     Update global model: $\theta_g^{t+1} \leftarrow \sum w_i \theta_i$.
20:     Accumulate Fisher: $F_g^{t+1} \leftarrow \sum w_i F_i$.
21: **end for**
22: **Return** Final model $\theta_g^T$.

---

directions, the model imposes tight constraints to strictly anchor to the global consensus; for low-curvature directions, it applies relaxed constraints to allow personalized adaptation (see Appendix D.1 and D.2 for geometric interpretation).

**(b) Curvature-aware Weighted Aggregation.** To construct a high-quality global model, during the server aggregation phase (Algorithm 1, Lines 19-22), the server performs weighted aggregation based on the Fisher Trace uploaded by clients. We regard the Fisher Trace $I_i = \text{Tr}(F_i)$ as the total amount of effective structural information contained in client $i$. The aggregation weight is calculated as $w_i = I_i / \sum_j I_j$, and the global state update rule is:

$$
\theta_g \leftarrow \sum_{i=1}^N w_i \theta_i, \quad F_g \leftarrow \sum_{i=1}^N w_i F_i \tag{8}
$$

This curvature-based aggregation strategy possesses superior noise resistance, automatically down-weighting clients with high gradient variance but low effective information (e.g., nodes containing numerous outlier noise graphs), thereby ensuring the stability of the global model under dual heterogeneous environments. The updated $\theta_g$ and $F_g$ are then broadcast to all clients as prior knowledge for the next round (Algorithm 1, Line 3) (see Appendix D.3 for geometric interpretation).

*Table 1.* Test accuracy comparison (Mean ± Std) on seven benchmarks. The best results are highlighted in **bold**.

| Method | DAPP | COLLAB | IMDB-BINARY | DD | MUTAG | NCI1 | PROTEINS |
|---|---|---|---|---|---|---|---|
| FedAvg | 74.00±2.15 | 67.36±1.88 | 81.40±1.25 | 81.41±2.45 | 83.51±4.12 | 73.11±1.67 | 67.12±2.05 |
| FedProx | 76.73±1.92 | 72.54±1.65 | 81.90±1.10 | 70.63±2.30 | 84.04±3.85 | 71.48±1.55 | 67.21±1.98 |
| Ditto | 76.23±1.45 | 77.14±1.42 | 80.20±0.98 | 68.25±2.10 | **85.11±3.50** | 71.95±1.32 | 67.65±1.75 |
| GCFL+ | 77.28±1.12 | 78.46±1.25 | 83.23±0.85 | 57.94±3.25 | 71.67±5.20 | 86.42±1.05 | 85.92±1.15 |
| FedStar | 75.24±1.35 | 76.54±1.30 | 81.56±0.92 | 59.44±2.88 | 58.33±6.50 | 80.00±1.22 | 55.03±2.45 |
| FedAGHN | 87.12±0.88 | 86.09±0.75 | 84.95±0.65 | **83.68±1.15** | 80.12±3.25 | 82.70±0.82 | 81.54±0.95 |
| SCFGL | 88.45±0.76 | 84.85±0.82 | 80.12±0.78 | 63.96±2.55 | 58.33±5.80 | 80.71±0.95 | 54.35±2.10 |
| **Ours** | **96.81±0.21** | **87.56±0.35** | **93.55±0.28** | 74.66±1.85 | 80.00±3.05 | **92.55±0.42** | **87.92±0.55** |

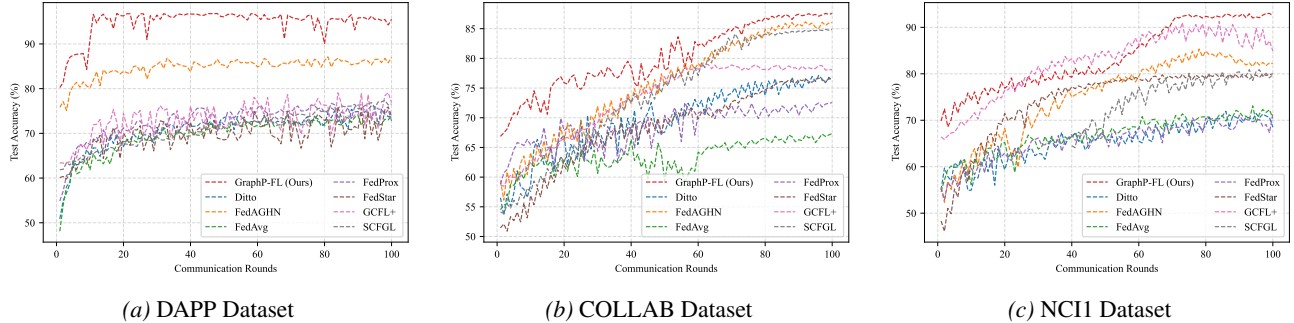

*(a)* DAPP Dataset     *(b)* COLLAB Dataset     *(c)* NCI1 Dataset

*Figure 3.* Test accuracy curves on DAPP, COLLAB, and NCI1 datasets during training.

## 5. Experiments

### 5.1. Experimental Setup

**Datasets.** We evaluate GraphP-FL on seven benchmarks across three domains: (1) **Bioinformatics:** NCI1, MUTAG, PROTEINS, and DD for molecular graph classification; (2) **Social Networks:** COLLAB and IMDB-BINARY for dense topological analysis; (3) **Encrypted Traffic:** DAPP, a large-scale real-world industrial dataset with extreme class imbalance and structural noise. This diverse selection rigorously tests model robustness across varying scales and complexities. Detailed statistics are provided in Table 6 in Appendix E.1 and E.3.

**Baselines.** We compare GraphP-FL against seven representative baselines across three categories: (1) **General FL: FedAvg** (McMahan et al., 2017) and **FedProx** (Li et al., 2020); (2) **Personalized FL: Ditto** (Li et al., 2021b); (3) **Federated Graph Learning (FGL): GCFL+** (Xie et al., 2021) (clustering-based), **FedStar** (Tan et al., 2023) (structure-task decoupling), **FedAGHN** (Song et al., 2025) (hypernetwork-based), and **SCFGL** (Fang et al., 2025) (structural entropy-based). These methods cover a broad range of strategies for addressing data and structural heterogeneity in FGL.

**Implementation Details.** We implement GraphP-FL using PyTorch on a standard workstation. We consistently apply a random 8:2 train-test split across all datasets, including both general benchmarks and DAPP. We adopt a 3-layer GraphCNN as the backbone. Crucially, we utilize Focal

Loss ($\gamma = 2.0$) for the imbalanced DAPP dataset to enhance minority class learning. All reported results are averaged over 5 independent runs with different random seeds. Detailed configurations (e.g., hyperparameters, hardware) are listed in Appendix E.2.

### 5.2. Main Performance Analysis

Table 1 presents the detailed accuracy comparison (Mean ± Std) between GraphP-FL and competitive baselines. GraphP-FL achieves SOTA performance on five out of seven benchmarks (DAPP, COLLAB, IMDB-BINARY, NCI1, and PROTEINS), demonstrating superior generalization.

**Superiority in Complex Heterogeneity.** GraphP-FL establishes a decisive advantage in challenging scenarios. On the large-scale industrial DAPP dataset and the topologically dense IMDB-BINARY, our method achieves significant gains of **8.36%** and **8.60%** over the best baseline, respectively. This stems from the adaptive denoising module, which effectively prunes task-irrelevant structural noise, enabling the extraction of robust discriminative features from noisy observations.

**Effectiveness of Fisher Alignment.** In bioinformatics, the Fisher elastic alignment proves its efficacy. On NCI1, GraphP-FL outperforms the clustering-based GCFL+ by **6.13%**. Notably, GraphP-FL maintains the lowest standard deviation across most datasets. This indicates that the anisotropic constraints provided by the Fisher Information Matrix effectively suppress stochastic oscillations during

*Table 2.* Robustness under different Non-IID settings ($\alpha$) on DAPP and NCI1.

| Heterogeneity ($\alpha$) | Dir(0.1) | | Dir(0.5) | | Dir(1.0) | |
|---|---|---|---|---|---|---|
| Method | DAPP | NCI1 | DAPP | NCI1 | DAPP | NCI1 |
| FedAvg | 73.85±2.55 | 70.52±2.10 | 74.00±2.15 | 73.11±1.67 | 74.48±1.55 | 73.48±1.45 |
| FedAGHN | 86.56±1.25 | 79.78±1.35 | 87.12±0.88 | 82.70±0.82 | 88.82±0.65 | 82.45±0.85 |
| **Ours** | **95.25±0.38** | **90.14±0.62** | **96.81±0.21** | **92.55±0.42** | **96.42±0.15** | **91.45±0.35** |

*Table 3.* Scalability analysis w.r.t Client Quantity ($N$) on DAPP and NCI1 datasets.

| Client Quantity ($N$) | Client=5 | | Client=10 | | Client=20 | |
|---|---|---|---|---|---|---|
| Method | DAPP | NCI1 | DAPP | NCI1 | DAPP | NCI1 |
| FedAvg | 74.00±2.15 | 73.11±1.67 | 73.48±2.45 | 71.58±1.95 | 72.33±2.88 | 70.28±2.25 |
| FedAGHN | 87.12±0.88 | 82.70±0.82 | 86.23±1.05 | 81.40±1.15 | 85.19±1.25 | 80.45±1.45 |
| **Ours** | **96.81±0.21** | **92.55±0.42** | **95.14±0.35** | **90.57±0.55** | **93.67±0.48** | **88.92±0.68** |

training, leading to significantly more stable convergence compared to the hypernetwork-based FedAGHN.

**Robustness in Small-sample Scenarios.** Regarding the performance fluctuations on DD and MUTAG, we attribute this to structural stability and small-sample bias. DD molecules possess high-certainty chemical bonds with low noise, saturating the gains from structural denoising. MUTAG contains only 188 graphs, where simple FedAvg benefits from its low-complexity bias. However, in stark contrast to the severe collapse of the structure-aware method FedStar on MUTAG (dropping to 58.33%), GraphP-FL maintains a robust performance of **80.00%**. This demonstrates that even when structural gains are limited, our Fisher regularization prevents catastrophic performance degradation.

## 5.3. Robustness and Scalability Analysis

We further analyzed the model's adaptability to extreme environments. Regarding robustness against statistical heterogeneity (Table 2), as Dirichlet $\alpha$ decreases from 1.0 to 0.1, all methods suffer performance degradation. However, GraphP-FL exhibits superior resilience. Notably on DAPP with $\alpha = 0.1$, our method maintains **95.25%** accuracy, significantly outperforming FedAGHN (86.56%).

Regarding scalability (Table 3), when the number of clients $N$ increases from 5 to 20, FedAGHN's accuracy on DAPP drops to 85.19% due to data sparsity. In contrast, GraphP-FL maintains **93.67%** accuracy, showing only a marginal drop (~3%), indicating high parameter efficiency.

## 5.4. Ablation Study

Table 4 presents the impact of removing specific components. (1) **w/o FRPA:** Removing Fisher alignment causes accuracy on DAPP to drop from 96.81% to 92.22%, indicating that the model fails to correct drift caused by Non-IID data without alignment. (2) **w/o GCL:** Removing the con-

trastive loss (accuracy drops to 93.69% on DAPP) deprives the model of intrinsic structural supervision. The superior performance of the full model confirms the complementarity of explicit alignment in parameter space and representation enhancement in feature space.

*Table 4.* Ablation study on component effectiveness. "Baseline" denotes the backbone encoder trained via FedAvg without specific regularization.

| Method | FRPA | GCL | DAPP | NCI1 | COLLAB |
|---|---|---|---|---|---|
| Baseline | × | × | 87.77 | 87.78 | 75.40 |
| w/o FRPA | × | ✓ | 92.22 | 89.77 | 79.64 |
| w/o GCL | ✓ | × | 93.69 | 90.56 | 85.45 |
| **Ours** | ✓ | ✓ | **96.81** | **92.55** | **87.56** |

## 5.5. Adaptive Hyperparameter Optimization

In this section, we conduct a comprehensive analysis of hyperparameter sensitivity to evaluate the performance of our method across diverse domains. While Section 5.4 demonstrated the necessity of the structural denoising constraint $\lambda_1$, we observe that applying a fixed $\lambda_1$ often leads to suboptimal performance due to severe domain heterogeneity. For instance, decentralized traffic networks (e.g., DAPP) inherently contain massive topological noise and thus demand strong denoising. Conversely, biochemical datasets (e.g., DD, MUTAG) possess highly pure and standardized topologies governed by strict physical laws, such as the maximum valence coordination number of carbon atoms in molecules and the stable folding architectures of proteins. Consequently, imposing excessive structural constraints on these specific domains risks destroying their intrinsic semantics.

To evaluate the adaptive capability of our method under this phenomenon, we design a dynamically adaptive tuning mechanism. By reparameterizing the constraint weights as learnable parameters $s_1$ and $s_2$, the objective function is

*Table 5.* Performance comparison using the adaptive hyperparameter mechanism.

| Model | DAPP | COLLAB | IMDB-B | DD | MUTAG | NCI1 | PROTEINS |
|---|---|---|---|---|---|---|---|
| FedAvg | $74.00 \pm 2.15$ | $67.36 \pm 1.88$ | $81.40 \pm 1.25$ | $81.41 \pm 2.45$ | $83.51 \pm 4.12$ | $73.11 \pm 1.67$ | $67.12 \pm 2.05$ |
| FedAGHN | $87.12 \pm 0.88$ | $86.09 \pm 0.75$ | $84.95 \pm 0.65$ | $83.68 \pm 1.15$ | $80.12 \pm 3.25$ | $82.70 \pm 0.82$ | $81.54 \pm 0.95$ |
| **Ours (Adapt)** | $\mathbf{95.79 \pm 0.43}$ | $\mathbf{87.86 \pm 0.33}$ | $\mathbf{92.43 \pm 0.54}$ | $\mathbf{82.56 \pm 0.88}$ | $\mathbf{86.25 \pm 0.47}$ | $\mathbf{92.75 \pm 0.53}$ | $\mathbf{90.27 \pm 0.65}$ |

automatically optimized based on domain-specific training feedback:

$$\mathcal{L}_{adapt} = \mathcal{L}_{sup} \exp(-s_1) + s_1 + \mathcal{L}_{SSL} \exp(-s_2) \\ + s_2 + \lambda_2 \mathcal{L}_{Riemann} \quad (9)$$

where the effective structural denoising weight becomes $\lambda_1^{(adapt)} = \exp(-s_2)$. The model autonomously balances the main classification task and the structural constraint by minimizing this joint objective during gradient descent.

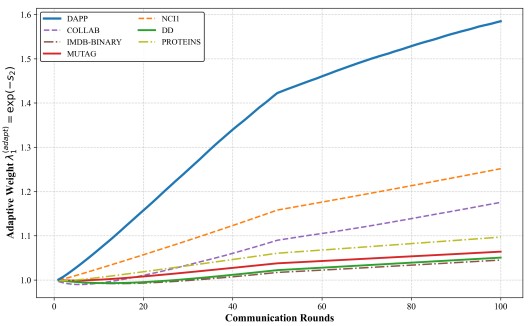

*Figure 4.* Cross-domain evolution of the adaptive structure denoising weight $\lambda_1^{(adapt)}$ during training.

As illustrated in Figure 4, starting from an initial $s_i = 0$ (i.e., $\lambda_1^{(adapt)} = 1.0$), the model achieves domain-specific adaptation. On the noisy DAPP dataset, the weight smoothly climbs to 1.58, actively enhancing noise filtering. Conversely, on pure datasets like DD and MUTAG, the weight stabilizes near 1.05, effectively protecting the original graph topology.

Consequently, as shown in Table 5, this adaptive mechanism yields remarkable performance gains. Compared to the fixed-weight baseline in Table 1, it improves accuracy on DD and MUTAG by 7.9% and 6.25% respectively. Interestingly, on the PROTEINS dataset, because the adaptive mechanism dynamically finds a more optimal constraint value, the performance is further improved by 2.35%, while maintaining prior excellent performance on the remaining datasets.

Besides superior accuracy, GraphP-FL is highly efficient and flexible in terms of communication, computation, and architectural design. We provide a detailed efficiency analysis evaluating model size and training time in Appendix E.4, alongside a comprehensive backbone compatibility and over-

head analysis in Appendix E.5. These evaluations collectively demonstrate that our method achieves the best trade-off between performance and computational cost.

## 6. Conclusion

In this paper, we proposed GraphP-FL, a general personalized FGL framework to address the dual heterogeneity of graph data. Specifically, the self-supervised dynamic topology reconstruction mechanism adaptively rectifies noisy structures, while the Fisher-based Elastic Parameter Alignment (FRPA) mechanism precisely protects local key knowledge. Extensive experiments on seven benchmarks demonstrate that GraphP-FL significantly outperforms state-of-the-art methods in complex heterogeneous scenarios. Future work will explore the potential of GraphP-FL in handling adversarial clients by leveraging Fisher information for anomaly detection.

## Acknowledgements

This work was supported by the National Natural Science Foundation of China under Grant No. 62572350, the Key Research and Development Program of Tianjin under Grant No. 23YFZCSN00240, and the Beijing-Tianjin-Hebei Natural Science Cooperation Project under Grant No. 25JJJJC0012. Additionally, the authors would like to thank the anonymous reviewers and Area Chairs for their constructive comments, which significantly improved the quality of this paper.

## Impact Statement

This paper presents work whose goal is to advance the field of Machine Learning. There are many potential societal consequences of our work, none which we feel must be specifically highlighted here.

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

# A. Detailed Graph Construction and Preprocessing Algorithms

### A.1. Alignment of Natural Graph Datasets

For natural graph data such as biochemical molecules (NCI1, PROTEINS, etc.) and social networks (COLLAB, IMDB-BINARY), raw data is typically stored in sparse matrix formats. We execute the following alignment steps:

- **Node Decoupling:** We utilize the graph index vectors provided by the datasets to map global nodes to independent graph instances $G_k$.

- **Feature Standardization:** For datasets containing discrete labels (e.g., COLLAB), we employ One-hot encoding to transform them into the node feature matrix $X$; for datasets containing continuous physical attributes (e.g., PROTEINS), we retain their original numerical features and apply Z-Score standardization.

- **Topology Preservation:** We directly retain the originally defined adjacency matrix $A$ without performing additional edge operations.

### A.2. Processing Details for DAPP Dataset

To handle unstructured encrypted traffic within the Federated Graph Learning framework, specifically utilizing the DAPP dataset (Shen et al., 2019), we propose a graph construction algorithm based on full-packet temporal topology. This algorithm transforms discrete traffic sessions into graph structural instances rich in semantics.

Given a bidirectional encrypted traffic session $S = \{p_1, p_2, \ldots, p_L\}$, where $p_t$ represents the $t$-th data packet ordered by timestamp, we formalize the graph construction process into the following three steps:

- **Node Instantiation:** We map every data packet $p$ in the flow to an independent node $v \in \mathcal{V}$ in the graph. Unlike traditional flow-statistical methods, our fine-grained mapping preserves the burst patterns of traffic at a microscopic scale.

- **Feature Space Mapping:** For each node, we construct a feature vector $x$ to encode the intrinsic attributes of the packet. Specifically, we extract the payload length of the packet as the core feature and perform normalization. To capture the interaction logic between the client and the server, we encode the transmission direction into the feature space: outbound packet lengths are mapped to positive values, and inbound packet lengths are mapped to negative values. Consequently, the node feature matrix $X \in \mathbb{R}^{L \times d_{in}}$ completely preserves the temporal fingerprint of the traffic.

- **Temporal Causal Edges:** To enable GNNs to capture the temporal context of traffic, we explicitly model the causal dependencies between packets in the edge set. We establish directed edges $(v_t, v_{t+1})$ between adjacent packet nodes, forming a main temporal chain. This chain-like topology makes the message passing mechanism of Graph Neural Networks equivalent to convolution operations in the temporal dimension, thereby effectively extracting sequence patterns during the encrypted handshake and data transmission phases.

Through the above method, we transform the raw DAPP traffic into a directed graph $G_{\text{DAPP}} = (X, A)$, enabling it to be directly processed by the adaptive denoising module of GraphP-FL.

# B. Derivation of Denoising Process (Section 4.2 (a))

To recover robust representations from noisy graph data $G_{obs} = (A_{obs}, X_{obs})$, we formalize the denoising process as a Variational Information Bottleneck (VIB) problem. According to the structural prior defined in Section 3.2, our goal is to learn a compressed representation $Z$ (i.e., the denoised graph) such that it satisfies:

$$\min_{\theta} \mathcal{L}_{IB} = \underbrace{-I(Z;Y)}_{\text{Maximizing Prediction}} + \beta \underbrace{I(Z; G_{obs})}_{\text{Minimizing Redundancy}} \tag{10}$$

Since the mutual information $I(Z; G_{obs})$ for high-dimensional graph data is intractable, we use variational inference to introduce a variational upper bound. Assuming $Z$ follows a parameterized distribution $p_\phi(Z|G_{obs})$, according to the variational inequality, we have:

$$I(Z; G_{obs}) \leq \mathbb{E}_{G \sim \mathcal{D}}[D_{KL}(p_\phi(Z|G)||r(Z))] \tag{11}$$

where $r(Z)$ is the prior marginal distribution (usually assumed to be a standard Gaussian). To achieve end-to-end gradient optimization, we model the generation process of $Z$ as a soft masking mechanism on the raw features. Specifically, we parameterize the posterior distribution $p_\phi(Z|G)$ as a deterministic mapping function $M_\Phi(\cdot)$, which learns the retention probability for each feature dimension.

This mechanism is mathematically equivalent to performing differentiable Bernoulli sampling on feature channels. By minimizing $\mathcal{L}_{IB}$, the model automatically forces the Mask values corresponding to dimensions with low mutual information with the downstream task $Y$ and high mutual information with the input $G_{obs}$ to tend towards 0. This effectively constructs a dynamic information gate that adaptively blocks the diffusion of noise during the GNN message passing process, thereby achieving indirect correction of the latent structure $A_{gt}$.

## C. Theoretical Analysis of Multi-scale Collaborative Optimization (Section 4.2 (b))

In this section, we provide theoretical support for the hierarchical graph encoding and self-supervised collaborative optimization strategy proposed in Section 4.2 (b) We first demonstrate the superiority of multi-scale aggregation in graph isomorphism discrimination, and then derive the theoretical connection between the contrastive loss $\mathcal{L}_{SSL}$ and mutual information maximization.

### C.1. Noise Resistance Analysis of Multi-scale Aggregation

**Proposition 1.** *Assuming the graph structure contains noise, and the feature representation of the l-th layer of GNN is expressed as the superposition of the true signal and propagation error: $h^{(l)} = h_{gt}^{(l)} + \epsilon_l$. Under the assumption of error variance accumulation with layers (Over-squashing of noise), there exists a set of attention weights $\{\alpha_l\}$ such that the error variance of the multi-scale fused representation $z = \sum_l \alpha_l h^{(l)}$ is strictly less than the error variance of the single-layer (using only the last layer) representation.*

**Analysis:** In Federated Graph Learning, structural noise cascades and amplifies as the number of GNN layers increases. For standard GIN, the output of the last layer $h^{(L)}$ often aggregates cumulative topological noise on the longest path, resulting in a large $Var(\epsilon_L)$. In contrast, shallow features $h^{(1)}$, although having a smaller receptive field, are less contaminated by topological noise ($Var(\epsilon_1)$ is smaller). The adaptive multi-scale fusion proposed in 4.2 (b) essentially performs inverse variance weighting. The attention mechanism $\alpha_l = \text{Softmax}(\text{Attn}(h^{(l)}))$ can dynamically assign weights based on feature purity. By assigning higher weights to low-noise shallow features, the model effectively constructs a structural smoother. This is equivalent to reducing the variance of the estimator by integrating views with different receptive fields, thus proving the theoretical superiority of this mechanism in combating the structural uncertainty described in Section 3.2.

### C.2. Adaptive Attention as Uncertainty Weighting

We introduced attention weights $\alpha_l = \text{Softmax}(\text{Attn}(h^{(l)}))$. From a Bayesian perspective, this is equivalent to inverse variance weighting of observations at different scales. Assume the representation $h^{(l)}$ of the $l$-th layer is a noisy estimate of the true latent semantic $z_{gt}$, i.e., $h^{(l)} = z_{gt} + \epsilon_l$, where noise $\epsilon_l \sim \mathcal{N}(0, \sigma_l^2)$. When a layer is severely affected by topological noise (e.g., deep layers aggregating erroneous long-distance neighbors), its estimation variance $\sigma_l^2$ increases. The attention mechanism essentially seeks the optimal linear unbiased estimate by learning the weights $\alpha_l$, i.e., assigning higher weights to layers with lower noise, thereby implicitly achieving smoothing of structural noise during the feature fusion stage.

### C.3. Derivation of Contrastive Loss and Mutual Information Lower Bound

In Section 4.2 (b), we claimed that the introduced self-supervised loss $\mathcal{L}_{SSL}$ instantiates a structural consistency prior, i.e., maximizing the mutual information $I(z_1; z_2)$ between augmented views. Here we provide the derivation process based on InfoNCE.

**Definition:** Let $z_1$ be the representation of the anchor view, $z_2$ be the representation of the positive sample view, and $\{z'\}$ be the set from the negative sample distribution. Our goal is to maximize the mutual information between $z_1$ and $z_2$:

$$I(z_1; z_2) = \sum_{z_1, z_2} p(z_1, z_2) \log \frac{p(z_1|z_2)}{p(z_1)} \tag{12}$$

According to the InfoNCE theory (Oord et al., 2018), mutual information has the following lower bound:

$$I(z_1; z_2) \geq \mathbb{E}\left[\log \frac{f(z_1, z_2)}{f(z_1, z_2) + \sum_{k=1}^{K} f(z_1, z'_k)}\right] + \log(K+1) \tag{13}$$

where $f(u, v) = \exp(\text{sim}(u, v)/\tau)$ is the density ratio estimation function, and $K$ is the number of negative samples. When we minimize $\mathcal{L}_{SSL}$ (i.e., NT-Xent loss) in the main text:

$$\mathcal{L}_{SSL} = -\log \frac{\exp(\text{sim}(z_1, z_2)/\tau)}{\sum_{z \in \{z_2, \mathcal{Z}'\}} \exp(\text{sim}(z_1, z)/\tau)} \tag{14}$$

We are actually maximizing the aforementioned mutual information lower bound. This means that through $\mathcal{L}_{SSL}$, we force the encoder to extract representations $z$ that maintain maximized mutual information even after different structural perturbations (such as denoising masks, random augmentations). In the context of probabilistic graphical models, this is equivalent to imposing a structural invariance prior, ensuring that the model captures the intrinsic topological semantics of the graph data that are independent of specific noise patterns.

## D. Theoretical Analysis of Fisher Elastic Collaboration (Section 4.3)

This section provides theoretical support for the Fisher-based elastic collaboration mechanism proposed in Section 4.3. We first derive the mathematical form of the Riemannian regularization term from a Bayesian perspective, then analyze how Fisher information encodes topological structural features in Graph Neural Networks, and finally provide a statistical interpretation for the trace-based aggregation weights.

### D.1. Derivation from KL Divergence to Riemannian Metric

In Section 4.3 (a), we introduced the regularization term $\mathcal{R}_{\text{Riemann}}(\theta_i, \theta_g) = \frac{1}{2}(\theta_i - \theta_g)^\top F_g(\theta_i - \theta_g)$. This form is not arbitrarily chosen but aims to constrain the Kullback-Leibler (KL) divergence between the local model distribution $P_{\theta_i}(Y|X)$ and the global prior distribution $P_{\theta_g}(Y|X)$.

On a statistical manifold, the natural distance between two parameterized distributions $P_\theta$ and $P_{\theta+\delta}$ is defined by the KL divergence. We consider the KL divergence $D_{KL}(P_{\theta_g}||P_{\theta_i})$ between the local parameters $\theta_i$ and the global anchor $\theta_g$. Assuming $\theta_i$ is in the neighborhood of $\theta_g$, we perform a second-order Taylor expansion of the KL divergence at $\theta_g$:

$$D_{KL}(P_{\theta_g}||P_{\theta_i}) \approx D_{KL}(P_{\theta_g}||P_{\theta_g}) + (\nabla_\theta D_{KL})^\top(\theta_i - \theta_g) + \frac{1}{2}(\theta_i - \theta_g)^\top H_{KL}(\theta_i - \theta_g) \tag{15}$$

where:

- The zeroth-order term $D_{KL}(P_{\theta_g}||P_{\theta_g}) = 0$.

- The first-order term vanishes because the KL divergence achieves its minimum (distance is 0) at $\theta_i = \theta_g$, so its gradient $\nabla_\theta D_{KL}|_{\theta=\theta_g} = 0$.

- The Hessian matrix $H_{KL}$ in the second-order term is exactly equal to the Fisher Information Matrix (FIM) $F_g$ at $\theta_g$.

According to the definition of KL divergence and the properties of the log-likelihood gradient, the elements of the Hessian matrix are:

$$H_{jk} = \frac{\partial^2}{\partial\theta_j \partial\theta_k} \mathbb{E}_x \left[ \sum_y P_{\theta_g}(y|x) \log \frac{P_{\theta_g}(y|x)}{P_\theta(y|x)} \right]\Bigg|_{\theta=\theta_g}$$

$$= \mathbb{E}_{x,y\sim P_{\theta_g}} \left[ -\frac{\partial^2 \log P_\theta(y|x)}{\partial\theta_j \partial\theta_k} \right] \tag{16}$$

$$= \mathbb{E}_{x,y\sim P_{\theta_g}} \left[ \nabla_\theta \log P_\theta (\nabla_\theta \log P_\theta)^\top \right] = F_g$$

Therefore, constraining the distribution distance $D_{KL} \leq \epsilon$ is equivalent under the second-order approximation to the constraint:

$$\frac{1}{2}(\theta_i - \theta_g)^\top F_g(\theta_i - \theta_g) \leq \epsilon \tag{17}$$

This proves that Eq. (7) in the main text is actually a distance constraint imposed on the Riemannian manifold (defined by the Fisher information metric $F_g$). This constraint considers the local curvature of the parameter space, aligning better with the geometric nature of probability distribution changes than the Euclidean distance $||\theta_i - \theta_g||^2$.

### D.2. Physical Meaning and Structural Encoding in GNNs

We further demonstrate that in the context of GNNs, the Fisher Information Matrix $F_g$ implicitly encodes topological structure information of the graph. Thus, FRPA is effectively a structure-aware parameter protection mechanism.

**Analysis:** Consider a standard Graph Convolutional layer (GCN), whose propagation formula is:

$$H^{(l+1)} = \sigma(\tilde{A}H^{(l)}W^{(l)}) \tag{18}$$

where $\tilde{A}$ is the normalized adjacency matrix, and $W^{(l)}$ is the weight parameter. The $k$-th diagonal element $F_{k,k}$ of the Fisher Information Matrix corresponds to the importance of parameter $W_k^{(l)}$, defined as the second moment of the gradient:

$$F_{k,k} = \mathbb{E} \left[ \left( \frac{\partial \mathcal{L}}{\partial W_k^{(l)}} \right)^2 \right] \tag{19}$$

According to the chain rule, the gradient of the loss function with respect to weight $W$ explicitly depends on the graph structure $\tilde{A}$:

$$\frac{\partial \mathcal{L}}{\partial W^{(l)}} = (\tilde{A}H^{(l)})^\top \frac{\partial \mathcal{L}}{\partial Z^{(l+1)}} \odot \sigma'(Z^{(l+1)}) \tag{20}$$

From this, it is evident that the magnitude of Fisher information $F$ is directly modulated by the adjacency matrix $\tilde{A}$:

- **High Connectivity Nodes / Key Motifs:** If certain parameters are responsible for processing key topological structures that appear frequently (such as Hub nodes in social networks or benzene rings in molecular graphs), the corresponding terms in $\tilde{A}$ will have large values, leading to large gradient magnitudes, and consequently large $F_{k,k}$.

- **Structure-Aware Constraint:** In FRPA, a larger $F_{k,k}$ produces a stronger constraint. This means the model automatically "locks" those parameters responsible for handling key topological structures, preventing them from being destroyed during local fine-tuning.

**Conclusion:** Unlike Fisher information in computer vision which often only encodes texture features, the Fisher matrix in GraphP-FL acts as a topological fingerprint, enabling the parameter alignment process to perceive and protect the underlying graph structural knowledge.

### D.3. Statistical Interpretation of Trace-based Aggregation Weights

In Section 4.3 (b), we adopt the Fisher trace $\mathrm{Tr}(F_i)$ as the aggregation weight. This can be interpreted as a high-dimensional generalization of Inverse-Variance Weighting.

Assume the local model parameters $\theta_i$ are estimates of the true global optimum $\theta^*$ and follow an asymptotic normal distribution:

$$\theta_i \sim \mathcal{N}(\theta^*, F_i^{-1}) \tag{21}$$

According to the Cramer-Rao lower bound, the inverse of the Fisher Information Matrix $F_i^{-1}$ approximates the covariance matrix (i.e., uncertainty) of the estimate. To obtain the global estimate with minimum variance, according to Bayesian fusion criteria, the optimal aggregation weights should be proportional to the precision matrix (i.e., $F_i$).

For computational feasibility, we use the trace $\mathrm{Tr}(F_i)$ as a scalar approximation of the matrix norm:

$$w_i \propto \text{Information Quantity}(\theta_i) \approx \mathrm{Tr}(F_i) \tag{22}$$

Therefore, giving higher weights to clients with high $\mathrm{Tr}(F_i)$ is essentially synthesizing the global model by maximizing global Fisher information (i.e., minimizing the uncertainty of the global estimate).

## E. Experimental Details and Visualization

### E.1. Dataset Statistics

We evaluate GraphP-FL on seven benchmark datasets spanning diverse domains: DAPP, NCI1, MUTAG, PROTEINS, DD, COLLAB, and IMDB-BINARY. Specifically, NCI1, MUTAG, PROTEINS, and DD are bioinformatics datasets consisting of chemical compounds and protein structures, used to assess standard molecular graph classification performance. COLLAB and IMDB-BINARY represent the social network domain, derived from scientific collaboration networks and actor interaction relationships, featuring dense topological structures. Finally, DAPP is a large-scale encrypted traffic dataset from a real-world industrial environment, characterized by extreme class imbalance and high-intensity structural noise, designed to rigorously test model robustness in realistic applications. Detailed statistics are provided in Table 6.

*Table 6.* Detailed statistics of the seven benchmark datasets used in experiments.

| Dataset | Category | Graphs | Classes | Avg. Nodes | Avg. Edges | Node Attr. | Edge Attr. |
|---|---|---|---|---|---|---|---|
| DAPP | Encrypted Traffic | 15,960 | 15 | 14.87 | 37.74 | Yes | No |
| COLLAB | Social Network | 5,000 | 3 | 74.49 | 2,457.78 | No | No |
| DD | Bioinformatics | 1,178 | 2 | 284.32 | 715.66 | Yes | No |
| IMDB-BINARY | Social Network | 1,000 | 2 | 19.77 | 96.53 | No | No |
| MUTAG | Bioinformatics | 188 | 2 | 17.93 | 19.79 | Yes | Yes |
| NCI1 | Bioinformatics | 4,110 | 2 | 29.87 | 32.30 | Yes | No |
| PROTEINS | Bioinformatics | 1,113 | 2 | 39.06 | 72.82 | Yes | No |

### E.2. Detailed Implementation

All experiments were conducted on an Intel Core i5-13600KF CPU (32GB RAM) using PyTorch 2.2. To ensure fair comparison, we adopt the GCFL-split strategy (Xie et al., 2021) for general benchmark datasets, while the large-scale DAPP dataset is randomly split with an 8:2 ratio. The model architecture employs a 3-layer GraphCNN as the backbone, with hidden dimensions set to 128 for DAPP and 64 for others. Clients use the Adam optimizer for local updates with an initial learning rate of 0.001, coupled with a StepLR scheduler (decay 0.5 every 50 rounds) for stable convergence. For the extremely imbalanced DAPP dataset, we employ Focal Loss ($\gamma = 2.0$), while standard Cross-Entropy Loss is used for others.

Regarding GraphP-FL specific configurations, the adaptive structure denoising module is consistently activated. The contrastive loss weight is set to 0.1 (temperature $\tau = 0.07$), and the trace of the Fisher Information Matrix is clipped within $[10^{-6}, 10^3]$ to prevent numerical instability. To ensure statistical reliability, all reported results are averaged over 5 independent runs with different random seeds, presented as mean test accuracy and standard deviation.

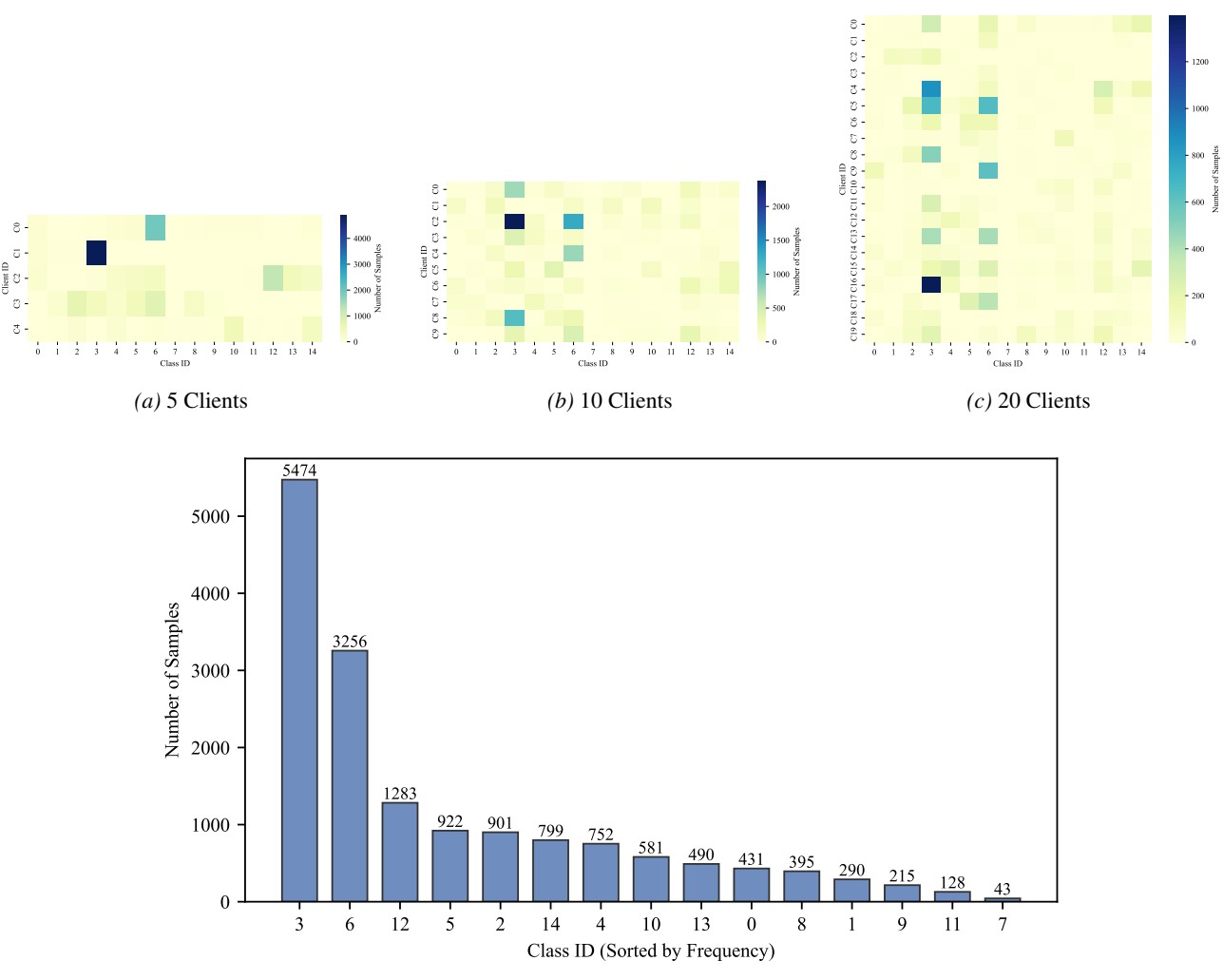

*(a)* 5 Clients          *(b)* 10 Clients          *(c)* 20 Clients

*(d)* Global Class Distribution (Long-tailed)

*Figure 5.* Visualization of dual heterogeneity on the DAPP dataset. **(a)-(c)** Heatmaps of local label distributions with client numbers $N \in \{5, 10, 20\}$ under Dirichlet $\alpha = 0.5$. The x-axis represents Class ID, and the y-axis represents Client ID; darker colors indicate more samples. Significant label absence and quantity disparity can be observed. **(d)** Global long-tailed distribution of classes, highlighting the extreme class imbalance problem.

### E.3. Data Heterogeneity Visualization

To visually demonstrate the statistical challenges in real-world industrial scenarios, we visualized the data distribution of the DAPP dataset under different client scales. As shown in Figure 5, the DAPP dataset exhibits significant dual imbalance characteristics:

- **Global Long-tailed Distribution:** As shown in Figure 5(d), the class distribution is extremely skewed. Head classes (e.g., Class 3) have over 5,000 samples, while tail classes (e.g., Class 7) have only a few dozen samples. This extreme inter-class imbalance severely challenges model fairness.

- **Local Non-IID:** As shown in the heatmaps in Figure 5(a)-(c), under a Dirichlet partition with $\alpha = 0.5$, the class distributions held by different clients vary hugely. Some clients possess samples from only a few classes, and the number of samples (indicated by color intensity) also differs significantly across clients. As the number of clients increases from 5 to 20, this data fragmentation and sparsity problem further intensifies.

### E.4. Efficiency Analysis

We validated the performance of GraphP-FL in terms of communication overhead and training time, as detailed in Table 7. **Regarding communication overhead,** GraphP-FL requires a model size of only 0.39 MB—approximately 50% of FedStar (0.78 MB), making it highly bandwidth-efficient. **Regarding computational efficiency,** although GraphP-FL incurs a slight per-round latency increase due to structure learning, it achieves dramatically faster convergence. Specifically, GraphP-FL reaches the optimal accuracy of 96.81% in just 23 rounds. In stark contrast, FedAvg struggles to converge, requiring 56 rounds to reach only 74.00% accuracy. Consequently, GraphP-FL reduces the total training time by 57% , demonstrating that our high-quality updates effectively minimize the communication rounds required for convergence.

*Table 7.* Computational and communication efficiency on DAPP.

| Method | Time/Round | Rounds | Total Time | Size |
|--------|-----------|--------|-----------|------|
| FedAvg | 19.4s | 56 | 18.11 min | 0.27 MB |
| GCFL+ | 16.3s | 89 | 26.02 min | 0.45 MB |
| FedStar | 18.8s | 65 | 26.10 min | 0.78 MB |
| **Ours** | 23.6s | **23** | **7.85 min** | **0.39 MB** |

### E.5. Backbone Compatibility and Overhead Analysis

To thoroughly evaluate the generalization capability and efficiency of the proposed framework, we conduct additional experiments by replacing the default backbone with other mainstream Graph Neural Networks, including GCN, GAT, GraphSAGE, and Graph Transformer.

As shown in Table 8, our framework consistently maintains high performance across all baseline architectures, demonstrating its strong compatibility. Notably, the default GraphP-FL configuration achieves the highest accuracy, justifying our architectural choice.

*Table 8.* Model performance across different GNN backbones.

| Model | DAPP | COLLAB | NCI1 |
|-------|------|--------|------|
| Ours + GCN | $91.62 \pm 0.65$ | $81.98 \pm 0.72$ | $91.88 \pm 0.58$ |
| Ours + GAT | $87.42 \pm 0.88$ | $84.49 \pm 0.91$ | $87.48 \pm 0.76$ |
| Ours + GraphSAGE | $92.54 \pm 0.71$ | $81.51 \pm 0.68$ | $88.80 \pm 0.82$ |
| Ours + Graph Transformer | $94.42 \pm 0.54$ | $81.98 \pm 0.82$ | $88.12 \pm 0.63$ |
| **GraphP-FL (Ours)** | $\mathbf{96.81 \pm 0.21}$ | $\mathbf{87.56 \pm 0.35}$ | $\mathbf{92.55 \pm 0.42}$ |

Furthermore, Table 9 details the computational overhead in terms of GPU Memory and Training Time per communication round. It is worth noting that while the Graph Transformer model demonstrates strong potential in centralized graph-level tasks and structural denoising, its global attention mechanism necessitates computing pairwise relationships between all node pairs. As illustrated in Table 9, this leads to a significant surge in computational overhead when processing large-scale dense graphs such as COLLAB, consuming up to 807.47MB of GPU memory and 39.67s per round.

*Table 9.* Overhead analysis (GPU Memory / Time per round).

| Model | DAPP | COLLAB | NCI1 |
|-------|------|--------|------|
| Ours + GCN | 35.73MB / 28.30s | 350.08MB / 7.73s | 37.82MB / 3.84s |
| Ours + GAT | 41.98MB / 29.53s | 1235.52MB / 12.06s | 51.87MB / 4.11s |
| Ours + GraphSAGE | 38.08MB / 27.02s | 201.35MB / 5.19s | 40.16MB / 3.71s |
| Ours + Graph Transformer | 115.12MB / 27.41s | 807.47MB / 39.67s | 26.26MB / 7.96s |
| **GraphP-FL (Ours)** | **42.87MB / 23.50s** | **216.63MB / 6.92s** | **47.43MB / 4.90s** |

This computational bottleneck is the fundamental reason why Graph Transformer was not adopted as the primary backbone in our main manuscript. In contrast, our default GraphP-FL maintains a highly lightweight footprint, requiring only 216.63MB

of memory and 6.92s on the COLLAB dataset, while achieving superior performance and demonstrating the best trade-off between performance and cost.

## F. Future Work

**Handling Adversarial Clients.** We recognize the significant potential of leveraging our Fisher-based elastic aggregation mechanism to address the challenge of adversarial clients. In scenarios where clients violate protocols due to malicious attacks or system failures—sending arbitrary messages such as data poisoning, model poisoning, or noisy parameters—our proposed **FRPA (Elastic Parameter Alignment)** framework demonstrates inherent defensive capabilities.

Rationale: Adversarial behaviors typically cause a client's Fisher information trace (Trace) or uploaded parameter distribution to exhibit statistically significant anomalies compared to the global consensus. Since Fisher information provides a precise quantification of parameter functional importance, these adversarial clients can be naturally identified as outliers within the Riemannian parameter manifold. Once identified, applying distribution-based weighted penalties to these nodes during the aggregation phase can effectively mitigate their importance scores. This approach minimizes their destructive impact on the global model aggregation process. Therefore, we envision that GraphP-FL holds substantial promise for resolving adversarial client issues. Enhancing the adversarial robustness of Federated Graph Learning is a critical and intriguing direction, and we leave the in-depth exploration of GraphP-FL in this domain for future work.

