# OpenReview forum: "GraphP-FL: Personalized Federated Graph Learning via Dynamic Structure Awareness and Fisher Information Elastic Alignment"
_ICML.cc/2026/Conference — ICML 2026 regular_

### Official Review · Reviewer_AhBq · 2026-03-06

**Soundness:** 3
**Presentation:** 3
**Significance:** 2
**Originality:** 3
**Overall Recommendation:** 3
**Confidence:** 4

**Summary:**

This paper proposes GraphP-FL, a novel personalized framework designed to address the "dual heterogeneity" challenge in Federated Graph Learning (FGL). This challenge manifests as structural noise (structural heterogeneity) in local graphs and model drift caused by Non-IID data distributions (statistical heterogeneity). The core contributions are twofold: (1) A dynamic topology reconstruction mechanism at the client side, which leverages self-supervised Graph Contrastive Learning (GCL) and the Variational Information Bottleneck (VIB) principle to adaptively denoise noisy graph structures even under label scarcity; and (2) A Fisher-based Regularized Parameter Alignment (FRPA) algorithm that introduces anisotropic regularization constraints in the parameter space. By quantifying parameter importance via the Fisher Information Matrix (FIM), FRPA aligns clients with the global model while preserving critical local knowledge, effectively mitigating catastrophic forgetting. Extensive experiments across seven benchmark datasets (spanning biochemistry, social networks, and encrypted traffic) demonstrate that GraphP-FL significantly outperforms state-of-the-art methods, achieving up to an 8.6% accuracy improvement, along with superior robustness and generalization capabilities.

**Compliance With Llm Reviewing Policy:**

Affirmed.

**Key Questions For Authors:**

Q1. Computational Efficiency
How does the overhead of Fisher estimation scale as the backbone GNN architecture becomes deeper or switches to a Graph Transformer?

Q2. Hyperparameter Adjustment
Table 7 only shows sensitivity for the DAPP dataset. Does the relative importance of λ1 (structure) and λ2 (alignment) require dynamic or domain-specific adjustment for the other 6 datasets?

Q3. Performance in Low-Noise Scenarios
On clean, small-scale datasets like DD and MUTAG, GraphP-FL shows marginal gains and even slightly underperforms FedAvg. The authors attribute this to "structural stability" and "small-sample bias." Is this due to the absence of structural noise, or does the method impose overly strong regularization in noise-free scenarios? Is there an adaptive mechanism to adjust the strength of structure learning based on the quality of the input graph?

**Limitations:**

The authors have provided a brief “Impact Statement” regarding societal consequences but have not adequately discussed the technical limitations of their work within the main text or conclusion. To improve the manuscript, the following constructive suggestions are offered:

1. Performance boundaries
Acknowledge that the method's advantages are clearest under significant noise/heterogeneity, and performance may plateau or be matched by simpler baselines on clean, small datasets.

2. Computational cost
Discuss the practical overhead of Fisher computation and potential mitigation strategies (e.g., periodic updates, approximations).

3. Potential misuse
Even if they see no immediate negative impact, discuss hypothetical risks (e.g., applying traffic classification to surveillance, or biased graph structures leading to unfair outcomes).

**Strengths And Weaknesses:**

Strengths:

S1. Soundness: The technical foundation is solid. The authors rigorously model structural noise as a latent variable inference problem and statistical heterogeneity as a prior constraint on a Riemannian manifold. Specifically, the use of the Fisher Information Matrix (FIM) to approximate KL divergence and construct anisotropic constraints is mathematically sound. The experimental design is comprehensive, covering seven diverse datasets, detailed ablation studies, and hyperparameter sensitivity analyses, which strongly support the central claims.

S2. Presentation: The paper is clearly written and well-structured. The narrative flows logically from problem definition to methodology and experimental validation. Figures (e.g., Figure 2 for the framework and Figure 4 for data heterogeneity visualization) are high-quality and aid understanding. The related work section provides a thorough categorization of existing FGL and PFL methods, clearly positioning the paper's contributions.

Weaknesses:

W1. Computational Overhead Details: Although Appendix E.4 mentions efficiency, the specific computational cost of calculating the diagonal approximation of the FIM on client devices (especially for large-scale graphs) is not fully elaborated. While diagonal approximation is used, the additional backpropagation required for Fisher estimation could introduce significant latency on resource-constrained edge devices, warranting further discussion.

W2. Hyperparameter Sensitivity: While sensitivity analysis is provided, the balance between the two main hyperparameters, λ1 (structure learning) and λ2 (Fisher alignment), appears to vary significantly across datasets (e.g., Table 7 only shows results for the DAPP dataset). It remains unclear if the relative importance of these modules requires dynamic adjustment across different domains.

W3. Limited Effectiveness in Low-Noise Scenarios: On clean, small-scale datasets like DD and MUTAG, GraphP-FL's advantages diminish—on DD it underperforms FedAvg (74.66% vs. 81.41%), and on MUTAG it merely ties with it. The authors attribute this to "structural stability" and "small-sample bias," but this reveals a deeper limitation: the method lacks an adaptive mechanism to detect when the input graph is already clean and reduce the strength of structure learning accordingly. Without such noise-awareness, the dynamic denoising module may over-regularize and introduce bias rather than remove it, potentially harming performance in precisely those scenarios where simplicity suffices. This boundary condition—where the method's complexity becomes a liability—is acknowledged but not systematically analyzed or addressed.

---

> ### Author Rebuttal · Authors · 2026-03-31
>
> We sincerely thank the reviewer for the valuable feedback. These constructive suggestions are of great help in improving our work. Our point-by-point responses to your questions are as follows.Full experimental results are available in the anonymous link: https://anonymous.4open.science/r/GraphP-FL-Tables3-0AE9/README.md
>
> 1.Key Question 1 & Weakness 1
>
> The Fisher mechanism in our framework is completely decoupled from the graph computational complexity of the backbone network, and it absolutely does not require any extra backpropagation process that you might be concerned about.
>
> First, as proven in Appendices D.1 and D.2, although the Fisher Information Matrix (FIM) represents the second-order curvature of the parameter space, we employ the diagonal approximation of the empirical Fisher (Eq. 18). In the underlying implementations of modern deep learning frameworks, when the local model completes the standard loss.backward(), the first-order gradient $g_k$ for each parameter already resides in the GPU memory. Our Fisher estimation simply executes extremely lightweight scalar squaring and moving average operations on these readily available gradients: $F_{k,k}=\mathbb{E}[g_k^2]$. Therefore, it eliminates the need for an additional data pass or an extra backpropagation cycle. Its computational complexity strictly converges to $O(d)$, where $d$ is the number of model parameters, which is completely independent of the topological scale of the graph.
>
> Second, as shown in Table 1 in the link, introducing the Fisher matrix alignment only incurs a minimal additional overhead of approximately 1.35MB in GPU memory and 1.18s per round. This is practically negligible on resource-constrained edge devices.
>
> By substituting the backbone network to validate the overhead, we observed that our scheme maintains a reasonable computational cost while achieving excellent classification accuracy, as shown in Tables 2 and 3 in the link. Notably, as the backbone network complexity increases, the overhead of the Fisher estimation remains consistently minimal, scaling at a constant $O(d)$. The actual computational bottleneck stems from the inherent graph-level operations of complex architectures themselves, which is precisely why we prioritized a lightweight architecture as our default solution.
>
> 2、Weakness 2 & Q2 & Weakness 3 & Q3
>
> We sincerely appreciate the reviewer’s profound insights regarding our performance on the DD and MUTAG datasets. As you accurately pointed out, these datasets are inherently clean, and excessive denoising inadvertently leads to information loss. Inspired by your constructive suggestions and literature [1,2], we have introduced a lightweight statistical scheme to pre-evaluate the dataset's noise scale and adaptively adjust the denoising constraints accordingly.
>
> First, by integrating domain prior knowledge, we statistically analyze node degrees to pre-assess graph purity. Empirical analysis shows that the DD protein dataset has an average degree of 5.03; in the MUTAG chemical dataset, the proportion of nodes with a degree of 5 or higher is strictly 0%. This is entirely determined by the physical connection principles of amino acids and carbon atoms, meaning these datasets contain almost no random redundant noise. Consequently, enforcing structural denoising inevitably leads to information loss, which explains why baselines with simpler structures temporarily held an advantage.
>
> Based on this evaluation, we conducted a sensitivity analysis on the denoising parameter $\lambda_1$ , as shown in Table 4 in the link. Under the default setting ($\lambda_1=0.1$), the model slightly trails the baselines. However, relaxing the denoising constraint yields performance gains of 1.77% and 2.83%. Conversely, enforcing a strong constraint leads to over-denoising, causing performance drops of 3.34% and 1.97%. This firmly corroborates the conclusion that inherently clean datasets do not require excessive denoising constraints.
>
> Finally, we conducted stress tests by injecting 20% random noise into the datasets, with results detailed in Table 5 in the link. Our model successfully achieved a 4%–5% performance improvement over its clean-set baseline. FedAvg and FedAGHN suffered a severe performance collapse on the noisy MUTAG dataset. This further demonstrates our model's advantage in noisy federated scenarios.
>
> Once again, we sincerely thank you for these invaluable modification suggestions. We will comprehensively incorporate these corresponding experiments and analyses into the revised manuscript.
>
> [1]Gnnguard: Defending graph neural networks against adversarial attacks. Advances in neural information processing systems.
>
> [2]Learning to drop: Robust graph neural network via topological denoising.Proceedings of the 14th ACM international conference on web search and data mining.

---

> > ### Author Rebuttal · Reviewer_AhBq · 2026-04-01
> >
> > Thank you for the detailed rebuttal and the additional experiments provided. While the responses address some of my initial concerns, there remain a few critical points that require further clarification:
> >
> > - To address the performance drop on low-noise datasets, the rebuttal introduces a heuristic based on average node degrees to pre-assess graph purity. This approach appears highly ad-hoc rather than a principled adaptive mechanism. How can a simple degree-based threshold reliably generalize to entirely different domains where dense graphs might be perfectly clean, or sparse graphs heavily corrupted?
> >
> > - The massive performance gains on the DAPP dataset rely on a specialized full-packet temporal topology construction algorithm. Can you explicitly confirm whether all competitive baselines were evaluated using this exact same graph construction? It is crucial to decouple the benefits of your proposed modules from the inherent advantages of this custom graph generation method.
> >
> > - The fundamental concern regarding hyperparameter sensitivity remains partially unaddressed. Given the vast topological differences between social networks, chemical molecules, and temporal traffic flows, does achieving the reported optimal performance strictly require manual, per-domain tuning of the balance between the structure learning constraint and the Fisher alignment?

---

> > > ### Author Response · Authors · 2026-04-07
> > >
> > > We sincerely thank the reviewer for the professional comments and constructive suggestions. We will now provide supplementary arguments and clarifications to address your core concerns point by point. Full experimental results and supplementary analysis are available in the anonymous link: https://anonymous.4open.science/r/GraphP-FL-Link3-B074/README.md
> > >
> > > Response to Questions (1) & (3):
> > >
> > > We fully agree with your perspective. The heuristic approach proposed in our previous response was indeed insufficiently rigorous to serve as a universal mechanism. Inspired by your constructive suggestions and the related work [1], we have designed a principled and dynamically adaptive hyperparameter tuning mechanism.
> > >
> > > First, we redefine the multi-task optimization objective. Let $\mathbf{W}$ be the model parameters, and $\sigma_1$ and $\sigma_2$ denote the observation noise parameters for the supervised classification task and the structure learning task, respectively. Our objective is to derive the maximum likelihood estimation of the joint Gaussian distribution. To facilitate network training, this process can be equivalently transformed into minimizing the following negative log-likelihood loss function:
> > >
> > > $$ L_{NLL}(W, \sigma_1, \sigma_2) \propto \frac{1}{2\sigma_1^2} L_{sup}(W) + \frac{1}{2\sigma_2^2} L_{SSL}(W) + \log(\sigma_1 \sigma_2) $$
> > >
> > > Next, to ensure numerical stability during backpropagation and avoid division-by-zero risks, we reparameterize the uncertainty parameters by defining learnable log-variances $s_i = \log(\sigma_i^2)$. By performing a change of variables on $\{L}_{NLL}$ and omitting the constant coefficients, the adaptive objective function of GraphP-FL is reconstructed as follows:
> > >
> > > $$ L_{adapt}(W, s_1, s_2) = L_{sup}(W) \exp(-s_1) + s_1 + L_{SSL}(W) \exp(-s_2) + s_2 + \lambda_2 L_{Riemann} $$
> > >
> > > The parameters $s_1$ and $s_2$ are updated automatically and synchronously with the network weights during the gradient descent optimization. Specifically, for graph datasets with highly pure topologies, such as DD and MUTAG, disrupting their structures strongly conflicts with the primary classification task, thereby triggering severe gradient fluctuations. Perceiving this feedback, the model automatically increases $s_2$ to exponentially reduce the weight of the implicit structural constraint, $\lambda_1^{(adapt)} = \exp(-s_2)$, thereby protecting the pure topology. Conversely, on highly noisy graphs like DAPP, structural denoising acts synergistically with the main task. It effectively filters out redundant interactions, resulting in an exceptionally smooth optimization process. Consequently, to minimize the $+s_2$ regularization penalty in the objective function, the model spontaneously maintains $s_2$ at a low level, thus preserving a strong denoising constraint.
> > >
> > > Finally, $\lambda_2\mathcal{L}_{Riemann}$ acts on the parameter space, not the data space. Leveraging the Fisher matrix's geometry for elastic curvature scaling, it requires no uncertainty weighting, fixing $\lambda_2=1.0$.
> > >
> > > As the linked Figure 1 shows, under the initial setting of $s_i=0$, the model achieves dynamic adaptation. On the noisy DAPP dataset, the model actively raises the constraint weight to 1.58 for denoising. Conversely, on pure datasets such as DD and MUTAG, the model stabilizes the constraint weight around 1.05 to avoid destroying the pure topological structures. As the linked Table 1 shows, on the DD and MUTAG datasets, since excessively strong denoising constraints are no longer imposed, the model realizes self-adaptation, resulting in performance improvements of 7.9% and 6.25% over the previous version, respectively. Interestingly, on the PROTEINS dataset, because the adaptive mechanism found a better constraint value, the performance is improved by 2.35% compared to the original, while maintaining the prior excellent performance on the other datasets.
> > >
> > > Response to Question (2):
> > >
> > > We clarify that raw DAPP data are unstructured sequences, requiring uniform conversion into graphs for all baseline models. As stated in Appendix A.2, the input graphs $G_{DAPP}$ for all baseline models are generated using the exact same mapping rules we proposed. Therefore, the performance superiority of GraphP-FL on the DAPP dataset is completely independent of the graph generation method itself.We will explicitly reiterate this fairness premise with prominent wording in the Section 5.1 of the revised manuscript to eliminate any misunderstandings.
> > >
> > > We sincerely thank you again for the valuable time and constructive feedback provided during the review process. Your suggestion has significantly enhanced the theoretical rigor of our work. We will incorporate all these modifications into the revised version of the manuscript.
> > >
> > > [1]Multi-task learning using uncertainty to weigh losses for scene geometry and semantics. CVPR.

---

### Official Review · Reviewer_Ndg2 · 2026-03-12

**Soundness:** 2
**Presentation:** 2
**Significance:** 3
**Originality:** 2
**Overall Recommendation:** 4
**Confidence:** 3

**Summary:**

This paper studies Federated Graph Learning (FGL), where multiple clients collaboratively train GNN while preserving data privacy. They propose GraphP-FL, a personalized federated graph learning framework that includes a self-supervised dynamic topology reconstruction mechanism and a Fisher-based elastic parameter alignment algorithm. Experiments on several datasets show that GraphP-FL outperforms state-of-the-art methods in most cases.

**Compliance With Llm Reviewing Policy:**

Affirmed.

**Final Justification:**

The rebuttal clarifies several key design choices and provides additional experimental evidence, which improves my understanding of the method. The added analysis on dataset characteristics and robustness strengthens the empirical justification. While some theoretical concerns remain only partially addressed, I find the overall contribution more convincing after the rebuttal. I therefore increase my score.

**Key Questions For Authors:**

Q1. Could the authors provide a clearer intuition for how the three loss terms achieve structural denoising and mitigate model drift?

Q2. Could the authors justify the approximations of the components in the proposed method and provide theoretical or empirical supports for these approximations? For example, variational information bottleneck and mutual-information-based objective are substituted with soft masking and NT-Xent, respectively.

Q3. The proposed method underperforms some baselines on the DD and MUTAG datasets. Could the authors provide additional analysis to better understand these results? For example, how does performance change under different numbers of clients or different coefficients for the loss terms?

**Limitations:**

yes

**Strengths And Weaknesses:**

Strength:

1. The paper introduces concepts from information theory, such as the information bottleneck, mutual information, and the Fisher information matrix, into federated graph learning. This provides an interesting perspective on topology reconstruction and parameter alignment.

2. The proposed method achieves strong empirical performance and outperforms existing methods on multiple datasets.

Weakness:

1. The paper provides limited intuitive explanation for why the combination of three terms in the loss objective is expected to achieve structural denoising or address the limitations of previous FGL methods.

2. The two newly introduced loss terms rely on several approximations without theoretical guarantees. For example, the topology reconstruction module is motivated by the variational information bottleneck but is implemented via soft masking. The mutual-information-based objective is substituted with NT-Xent. The Fisher regularized parameter alignment adopts a diagonal approximation.

3. The proposed method underperforms some baselines on the DD and MUTAG datasets, but the paper does not provide further experiments and analysis of these cases.

---

> ### Author Rebuttal · Authors · 2026-03-31
>
> We sincerely thank the reviewer for the valuable feedback. These constructive suggestions are of great help in improving our work. Our point-by-point responses to your questions are as follows.Full experimental results and supplementary analysis are available in the anonymous link: https://anonymous.4open.science/r/GraphP-FL-Tables2-73B7/README.md
>
> 1.Key Question 1 & Weakness 1
>
> The three loss terms are designed to collaboratively address the model drift issue in federated graph learning, which is driven by high topological noise and high data heterogeneity.
>
> The supervised loss $\mathcal{L}_{sup}$ aims to guide the model to fit the local data distribution of clients, maintaining the fundamental classification functionality.
>
> $\mathcal{L}_{SSL}$ maximizes the mutual information between two different views, enabling the model to memorize the core structure of the graph data while filtering out noisy and redundant edges, thereby achieving a denoising effect. Furthermore, we provide a detailed experimental analysis of this mechanism in our response to Key Questions 3 and Weaknesses 3.
>
> $\mathcal{R}_{Riemann}$ provides elastic protection and knowledge alignment in the parameter space. By utilizing the FIM matrix to measure the contribution of each parameter to local structural knowledge, it can accurately identify critical parameters and assign protection weights, effectively preventing catastrophic forgetting.
>
> 2.Key Questions 2 and Weaknesses2
>
> The three approximations utilized in our framework are not heuristic designs, but are supported by rigorous mathematical bounding guarantees. Due to the space constraints of this rebuttal, we respectfully request the reviewer to refer to the Appendices of the main manuscript for the detailed mathematical derivations.
>
> Variational Information Bottleneck (VIB) and Soft Masking (Appendix B)
>
> Mutual Information and NT-Xent (Appendix C.3)
>
> Diagonal Approximation of Fisher Regularization (Appendix D.1)
>
> 3.Key Questions 3 and Weaknesses3
>
> We sincerely apologize for the brief explanation in the paper regarding the performance on the DD and MUTAG datasets. We clarify that our method not achieving absolute dominance on these two datasets is not a flaw of the framework itself.
>
> First, our statistical analysis reveals that the DD protein dataset has an average node degree of 5.03 and a maximum degree of 19. In the MUTAG chemical dataset, the proportion of nodes with a degree of 5 is exactly 0%. This is strictly dictated by the real-world connection principles of amino acids and carbon atoms. Consequently, these two datasets are inherently clean and contain almost no random redundant noise. Enforcing structural denoising on such noise-free graphs inevitably leads to information loss, which allows baseline models with simpler structures to maintain a slight edge. Naturally, such datasets do not require overly strong denoising constraints.
>
> To validate our conclusion, we conducted experiments by augmenting the datasets with 20% random noise. As shown in Table 1 (available in the provided link), our model’s performance improved by 4%–5% compared to its original results. In stark contrast, baseline models such as FedAvg and FedAGHN experienced a performance collapse on the noisy MUTAG dataset.
>
> Secondly, we conducted a sensitivity analysis on the parameter $\lambda_1$ of the denoising module, as shown in Table 2 in the link. When we set $\lambda_1 = 0.1$, the model's performance is indeed inferior to some baseline models. However, when we set $\lambda_1 = 0.01$ to reduce the constraint of the denoising module, our model achieves performance improvements of 1.77% and 2.83% by avoiding excessive denoising. When we set $\lambda_1 = 0.5$ to apply a strong constraint, the model over-denoises the clean dataset and mistakenly removes objectively existing real edges, resulting in performance decreases of 3.34% and 1.97%. This further validates the conclusion that excessively strong constraints are unnecessary for clean datasets.
>
> Finally, we evaluate the scalability of our model on these datasets, as shown in Table 3 in the link. As the number of clients increases from 5 to 20, the accuracy of FedAvg and FedAGHN degrades, with a maximum drop of nearly 9%. In contrast, our framework maintains stable accuracies of 75.36% on DD and 81.11% on MUTAG under the N=20 setting, consistently outperforming all baseline models. This demonstrates that simple-structured baseline models may achieve high scores via direct aggregation under ideal data distribution scenarios, but suffer from performance drift in extremely heterogeneous scenarios.
>
> We sincerely thank you again for your valuable and constructive suggestions. We will incorporate these detailed privacy and security analyses into the revised manuscript.

---

> > ### Author Rebuttal · Reviewer_Ndg2 · 2026-04-04
> >
> > Thank you for the detailed rebuttal. The clarifications are helpful, and I encourage the authors to incorporate them into the final version. I will increase my scores.

---

> > > ### Author Response · Authors · 2026-04-07
> > >
> > > Response to Reviewer Acknowledgement:
> > >
> > > We sincerely thank you for your time, your constructive feedback throughout the review process, and your positive reassessment of our work. We are very glad that our detailed rebuttal has fully addressed your concerns. As suggested, we will certainly incorporate all the detailed clarifications and revisions into the final version of the manuscript. Thank you again for helping us improve our paper.
> > >
> > > Best regards,
> > >
> > > The Authors

---

### Official Review · Reviewer_8xHN · 2026-03-12

**Soundness:** 2
**Presentation:** 3
**Significance:** 3
**Originality:** 3
**Overall Recommendation:** 4
**Confidence:** 4

**Summary:**

This paper addresses the dual heterogeneity challenge in graph-level FGL: structural noise within local graphs and statistical heterogeneity across different clients. For structural noise, the framework employs a self-supervised dynamic topology reconstruction mechanism based on the Variational Information Bottleneck to denoise local graphs and enhance the representation learning. For model drift and catastrophic forgetting, Fisher-based Elastic Parameter Alignment module is introduced to adaptively constraint the parameter in a manifold to align with the global model. Furthermore, the server aggregates models using a curvature-aware weighting scheme based on Fisher Trace.

**Compliance With Llm Reviewing Policy:**

Affirmed.

**Final Justification:**

Considering the content of the author's rebuttal, I would keep my current positive rating.

**Key Questions For Authors:**

How could this framework adapt to other GNN architectures?

How about using models or methods regarding the GraphTransformer or transformer-based architecture?

**Limitations:**

See weakness 4

**Strengths And Weaknesses:**

Strengths:

1.	The proposed method considers both statistical heterogeneity and imperfect structure issues in FGL setting, which is beyond previous works.
2.	The designs are reasonable for structure and statistics heterogeneity. VIB for denoising, multi-scale representation and SSL for better representation learning, Fisher-based regularization on Riemannian Manifold and aggregation for statistical non-iid, they look sound and have physical explanation.
3.	The experiments cover a wide range of real-world graph datasets across 3 domains. Some are inherently graph data and some need to construct graphs. The proposed method is also communication efficient.

Weaknesses:

1.	Additional Complexity: VIB, multi-scale representation fusion and self-supervised learning modules introduce additional complexity. Computing fisher matrix needs high-order gradients. An analytical and empirical analysis of the overhead (computation and memory footprint) is needed.
2.	It’s not clear whether the proposed framework could support other GNN models. In the experiment, GraphCNN is used and GIN is adopted for multi-scale representation fusion. Whether the proposed framework supports other architectures such as GAT, GraphSAGE...?
3.	For the graph-level task and structure noises, transformer-based models are choices (also discussed in the paper). But the main content of the paper does not discuss or compare with these methods.
4.	Privacy concerns on uploading the Fisher diag: The local Fisher Information Matrix inherently reveals the topology of the client's data. Since GNN weight gradients are directly multiplied by the normalized adjacency matrix during backpropagation (Eq. 19), the FIM directly encodes this structural dependence. Consequently, uploading the FIM diagonal inadvertently exposes sensitive local topological features.

---

> ### Author Rebuttal · Authors · 2026-03-31
>
> We sincerely thank the reviewer for the valuable feedback. These constructive suggestions are of great help in improving our work. Our point-by-point responses to your questions are as follows.Full experimental results and supplementary analysis are available in the anonymous link: https://anonymous.4open.science/r/GraphP-FL-Tables1-CD80/README.md
>
> 1. Key Question 1 and Weakness 2
>
> By substituting the backbone network, we validated the adaptability of our proposed model across various Graph Neural Network (GNN) architectures, consistently achieving excellent classification accuracy. Furthermore, even when processing the large-scale dense graph COLLAB, our scheme successfully maintains computational overhead within a reasonable range, as illustrated in Tables 1 and 2 in the provided link.
>
> 2.Key Question 2 and Weaknesses 3
>
> The Graph Transformer model demonstrates strong potential in centralized graph-level tasks and structural denoising. However, as its global attention mechanism necessitates computing pairwise relationships between all node pairs, it leads to a significant surge in computational overhead when processing large-scale dense graphs such as COLLAB (as illustrated in Tables 3 and 4 in the link). This bottleneck is the fundamental reason why it was not adopted as the primary backbone in our main manuscript.
>
> 3.Weakness 1
>
> Through our experimental analysis, the VIB and multi-scale fusion modules incur a minimal per-round time overhead of 0.71s. Due to the necessity of multi-view forward propagation, the self-supervised learning module increases the GPU memory footprint by approximately 14.7MB and the time overhead by 2.18s, as detailed in Table 5 in the provided link.
>
> The concern you raised regarding high-order gradient issues with the Fisher Information Matrix (FIM) may stem from our mention of the Hessian matrix of the KL divergence in the derivation in Appendix D.1. However, as proven in Eq. 15 of Appendix D.1, the Hessian of the KL divergence is theoretically equivalent to the covariance of the log-likelihood gradients. Correspondingly, in Eq. 18 of Appendix D.2, we employ the empirical Fisher with a diagonal approximation mechanism: $F_{k,k} = \mathbb{E}[(\frac{\partial \mathcal{L}}{\partial w_k^{(l)}})^2]$. During local training on the clients, we exclusively rely on the first-order gradients generated by standard backpropagation, taking their squares and calculating moving averages. This approximation successfully reduces the computational complexity from $O(d^2)$ to $O(d)$, thus avoiding any additional high-order computational overhead.
>
> 4.Weakness 4
>
> First, we appreciate the reviewer’s consideration regarding privacy and security. Your point that GNN gradients are correlated with the adjacency matrix (as shown in Eq. 19) is entirely correct. However, uploading the diagonal approximation of the local Fisher Information Matrix (diag FIM) does not lead to the leakage of sensitive local topological features. The local adjacency matrix resides in a variable-length discrete space of $O(N^2)$, whereas the FIM diagonal exists in a fixed-length continuous parameter space of $O(d)$. As shown in Eq. 18 and 19, the calculation of the FIM involves global summation across all nodes, squaring operations, and statistical averaging across batches. This dimension reduction mapping is highly non-injective and involves substantial information loss. Consequently, it is impossible for an attacker to reverse-engineer specific microscopic edges or node degrees from these fixed-length parameter-level scalars.
>
> Furthermore, the message-passing mechanism of GNNs strictly satisfies permutation invariance. Therefore, countless isomorphic topologies or heterogeneous topologies that yield equivalent feature aggregation results will map to the exact same gradients. Thus, the FIM diagonal only encodes macroscopic parameter structural sensitivity and cannot reflect the actual entity-level edge relationships within the graph data.
>
> We sincerely thank you again for your valuable and constructive suggestions. We will incorporate these detailed privacy and security analyses into the revised manuscript.

---

> > ### Author Rebuttal · Reviewer_8xHN · 2026-04-02
> >
> > Thanks for the authors' rebuttal.
> >
> > As said, it's still suggested to include more discussion about the cost and privacy issues. The author provides empirical experiments showing that the additional compute and GPU footprint overheads are marginal. However, it's likely because the graphs in current graph-level datasets are small (e.g., avg 284 nodes in DD). For privacy, although the batching used in training would confuse the FIM, in highly non-IID scenarios, a client would possess local graphs (mostly) belonging to the same class. Statistical averaging across batches would still reveal some common patterns of this class.
> >
> > Overall, I will keep my positive rating.

---

> > > ### Author Response · Authors · 2026-04-07
> > >
> > > Response to Reviewer Acknowledgement:
> > >
> > > We sincerely thank you for your continued support and for maintaining your positive rating! Your concerns regarding the scalability on larger graphs and the privacy risks in extreme non-IID scenarios are highly accurate. We would like to discuss these aspects as follows:
> > >
> > > Regarding cost and scalability in large-scale data scenarios, the computational complexity of our GNN backbone and self-supervised denoising module scales linearly with the number of nodes $|V|$ and edges $|E|$, i.e., $\mathcal{O}(|V| + |E|)$. To reduce computational complexity, the global Fisher matrix uses a diagonal approximation, drastically reducing the complexity from $\mathcal{O}(d^2)$ to linear $\mathcal{O}(d)$. Therefore, the additional time overhead introduced by computing the Fisher information depends entirely on the model parameter size and is completely independent of the graph size. This means that even in massive graph scenarios, the computational overhead will only scale linearly without uncontrollable surges. Furthermore, our designed elastic alignment mechanism significantly accelerates training convergence, which drastically reduces the total number of required global communication rounds and lowers overall training costs, further ensuring its high practicality in large-scale scenarios.
> > >
> > > Regarding the potential privacy risks in extreme non-IID scenarios, we fully agree that this must be taken seriously in the design, just as you pointed out. In our aggregation mechanism, clients only need to upload their model parameters $\theta_i$, the diagonal Fisher matrix $F_i$, and a single scalar $I_i$ per communication round. Since $F_i$ is a diagonal matrix, its dimension strictly matches the parameter size, meaning the communication payload per round is only $\mathcal{O}(|\theta|)$. It is precisely because our aggregation mechanism requires such a compact model to transmit that it is naturally highly suitable for integration with technologies like Secure Multi-Party Computation or Homomorphic Encryption, thereby resolving privacy leakage issues at a very low overhead.
> > >
> > > As suggested, we will explicitly add a dedicated discussion section in the final manuscript to address these specific cost scalability and privacy considerations. Thank you again for your rigorous and constructive guidance throughout the review process!
> > >
> > > Best regards,
> > >
> > > The Authors

---

### Decision · Program_Chairs · 2026-04-30

**Decision:**

Accept (regular)

**Comment:**

Reviewers are generally positive on the method, but some issues were raised with both the datasets selected, and theoretical justifications for the method.

Strengths:
- proposed method considers both statistical heterogeneity and imperfect structure issues in FGL
- interesting perspective on topology reconstruction and parameter alignment.
- strong empirical performance and outperforms existing methods on multiple datasets.

Weaknesses:
- Evaluated primarily on older, small, datasets (including ones with known problems like MUTAG/DD/NC1).  Proposed method showed inconsistent performance on these.
- balance between the two main hyperparameters, λ1 (structure learning) and λ2 (Fisher alignment), appears to vary significantly across datasets